# Establishment of developmental gene silencing by ordered polycomb complex recruitment in early zebrafish embryos

Graham JM Hickey[1], Candice L Wike[1], Xichen Nie[1], Yixuan Guo[1], Mengyao Tan[1], Patrick J Murphy[1,2], Bradley R Cairns[1]*

[1]Howard Hughes Medical Institute, Department of Oncological Sciences and Huntsman Cancer Institute, University of Utah School of Medicine, Salt Lake City, United States; [2]Department of Biomedical Genetics, Wilmot Cancer Center, University of Rochester School of Medicine, Rochester, United States

**Abstract:** Vertebrate embryos achieve developmental competency during zygotic genome activation (ZGA) by establishing chromatin states that silence yet poise developmental genes for subsequent lineage-specific activation. Here, we reveal the order of chromatin states in establishing developmental gene poising in preZGA zebrafish embryos. Poising is established at promoters and enhancers that initially contain open/permissive chromatin with 'Placeholder' nucleosomes (bearing H2A.Z, H3K4me1, and H3K27ac), and DNA hypomethylation. Silencing is initiated by the recruitment of polycomb repressive complex 1 (PRC1), and H2Aub1 deposition by catalytic Rnf2 during preZGA and ZGA stages. During postZGA, H2Aub1 enables Aebp2-containing PRC2 recruitment and H3K27me3 deposition. Notably, preventing H2Aub1 (via Rnf2 inhibition) eliminates recruitment of Aebp2-PRC2 and H3K27me3, and elicits transcriptional upregulation of certain developmental genes during ZGA. However, upregulation is independent of H3K27me3 – establishing H2Aub1 as the critical silencing modification at ZGA. Taken together, we reveal the logic and mechanism for establishing poised/silent developmental genes in early vertebrate embryos.

*For correspondence:
brad.cairns@hci.utah.edu

Competing interest: The authors declare that no competing interests exist.

## Editor's evaluation

This manuscript uses genomics tools and pharmacological treatment to study the chromatin landscape change in early-stage Zebrafish embryos, at the critical stage from using maternally deposited transcripts to actively turning on embryotic gene expression. In particular, this work addresses how key chromatin factors coordinate in regulating distinct groups of genes during early vertebrate development and should be of interest to researchers in chromatin biology and developmental biology fields.

## Introduction

Early vertebrate embryos initiate embryonic/zygotic transcription, termed zygotic genome activation (ZGA), and must distinguish active housekeeping genes from developmental genes, which must be temporarily silenced, but kept available for future activation (*Chan et al., 2019*; *Lindeman et al., 2011*; *Murphy et al., 2018*; *Potok et al., 2013*; *Vastenhouw et al., 2010*). Developmental gene promoters in early embryos are packaged in 'active/open' chromatin – which can involve a combination of histone variants (e.g. H2A.Z), open/accessible chromatin (via ATAC-seq), permissive histone modifications (e.g. H3K4me1/2/3, H3K27ac), and (in vertebrates such as zebrafish) focal DNA hypomethylation (*Akkers et al., 2009*; *Bogdanovic et al., 2012*; *Chan et al., 2019*; *Jiang et al., 2013*; *Lindeman*

*et al., 2011*; *Murphy et al., 2018*; *Potok et al., 2013*; *Vastenhouw et al., 2010*). As H3K9me3 and H3K27me3 are very low or absent at ZGA in zebrafish, it remains unknown how developmental gene silencing occurs at ZGA within an apparently permissive chromatin landscape, and how subsequent H3K27me3 is established at developmental genes during postZGA stages.

We addressed these issues further in zebrafish, which conduct full ZGA at the tenth synchronous cell cycle of cleavage stage (~3.5 hr post fertilization (hpf), ~ 2000 cells) (*Schulz and Harrison, 2019*). Prior to ZGA (preZGA), zebrafish package the promoters and enhancers of housekeeping genes and many developmental genes with chromatin bearing the histone variant H2afv (a close ortholog of mammalian H2A.Z, hereafter termed H2A.Z(FV)), and the 'permissive' modifications H3K4me1 and H3K27ac (*Murphy et al., 2018*; *Zhang et al., 2018*); a combination termed 'Placeholder' nucleosomes – as they hold the place where poising/silencing is later imposed (*Murphy et al., 2018*). Curiously, at ZGA in zebrafish (and also in mice and humans), silent developmental gene promoters also contain H3K4me3, a mark that normally resides at active genes (*Dahl et al., 2016*; *Lindeman et al., 2011*; *Liu et al., 2016*; *Vastenhouw et al., 2010*; *Xia et al., 2019*; *Zhang et al., 2016*). After ZGA, developmental genes progressively acquire H3K27me3 via deposition by polycomb repressive complex 2 (PRC2) (*Lindeman et al., 2011*; *Liu et al., 2016*; *Vastenhouw et al., 2010*; *Xia et al., 2019*). Here, we address the central issues regarding how developmental genes bearing Placeholder nucleosomes and H3K4me3 are transcriptionally silenced during preZGA and ZGA stages in the absence of H3K27me3, and how subsequent H3K27me3 is focally established during postZGA (*Figure 1—figure supplement 1A*).

## Results

### H2Aub1 is present in preZGA zebrafish embryo chromatin

First, we sought a repressive histone modification that might explain how developmental genes are silenced at ZGA. We examined zebrafish embryos at preZGA (2.5 hpf, ~256 cells), ZGA (3.5 hpf ~2000 cells), and postZGA (4.3 hpf, >4 K cells) – and confirmed very low-absent H3K27me3 (*Figure 1—figure supplement 1B*; *Lindeman et al., 2011*; *Vastenhouw et al., 2010*) and H3K9me3 absence (*Laue et al., 2019*) during preZGA and ZGA, but revealed the presence of histone H2A monoubiquitination at lysine 119 (termed hereafter H2Aub1), a repressive mark deposited by the polycomb repressive complex 1 (PRC1) (*Figure 1A and B*, one of three biological replicates is displayed) (*de Napoles et al., 2004*; *Kuroda et al., 2020*; *Wang et al., 2004b*). Notably, zebrafish sperm lacked H2Aub1 whereas oocytes displayed H2Aub1 (*Figure 1B*; *Figure 1—figure supplement 1C*; *Figure 1—figure supplement 2A*). Current antibodies were not designed to distinguish H2A.Z(FV)ub1 from H2Aub1, so hereafter we refer to the epitope as H2Aub1.

### Developmental promoters acquire Placeholder, Rnf2, and H2Aub1 during preZGA

To localize H2Aub1 we conducted chromatin immunoprecipitation (ChIP) experiments at preZGA, ZGA, and postZGA (replicate structures for *Figure 1—figure supplement 2B-D*), and examined promoters (*Figure 1*) and enhancers (*Figure 2*). For all ChIP experiments, two to three biological replicates were conducted, which involved isolating different batches of zebrafish embryos. To more finely examine ZGA, we conducted additional ChIP profiling of Placeholder nucleosomes at ZGA (3.5 hpf), which complemented our prior profiling at preZGA (2.5 hpf) and postZGA (4.3 hpf) (*Murphy et al., 2018*) (replicate structure for *Figure 1—figure supplement 2E, F*). Interestingly, we found H2Aub1 highly co-localized at gene promoters and enhancers with Placeholder nucleosomes, H3K27ac (*Zhang et al., 2018*; *Figure 1C*), and ATAC-seq sensitive chromatin (*Figure 1—figure supplement 2L*, two biological replicates) during preZGA and ZGA stages. However, during postZGA, high levels of H2Aub1 overlap with only a portion of Placeholder-bound loci, an observation explored further, below. For our comparisons to DNA methylation (DNAme), we note that DNAme patterns are reprogrammed between fertilization and the preZGA (2.5 hpf) timepoint (*Potok et al., 2013*; *Jiang et al., 2013*), but remain static in zebrafish embryos from 2.5 hpf (preZGA) to 4.3 hpf (postZGA). Therefore, for brevity we chose to display only a single timepoint for DNAme data in subsequent figures, which is representative of all developmental stages examined by the genomics approaches in this work.

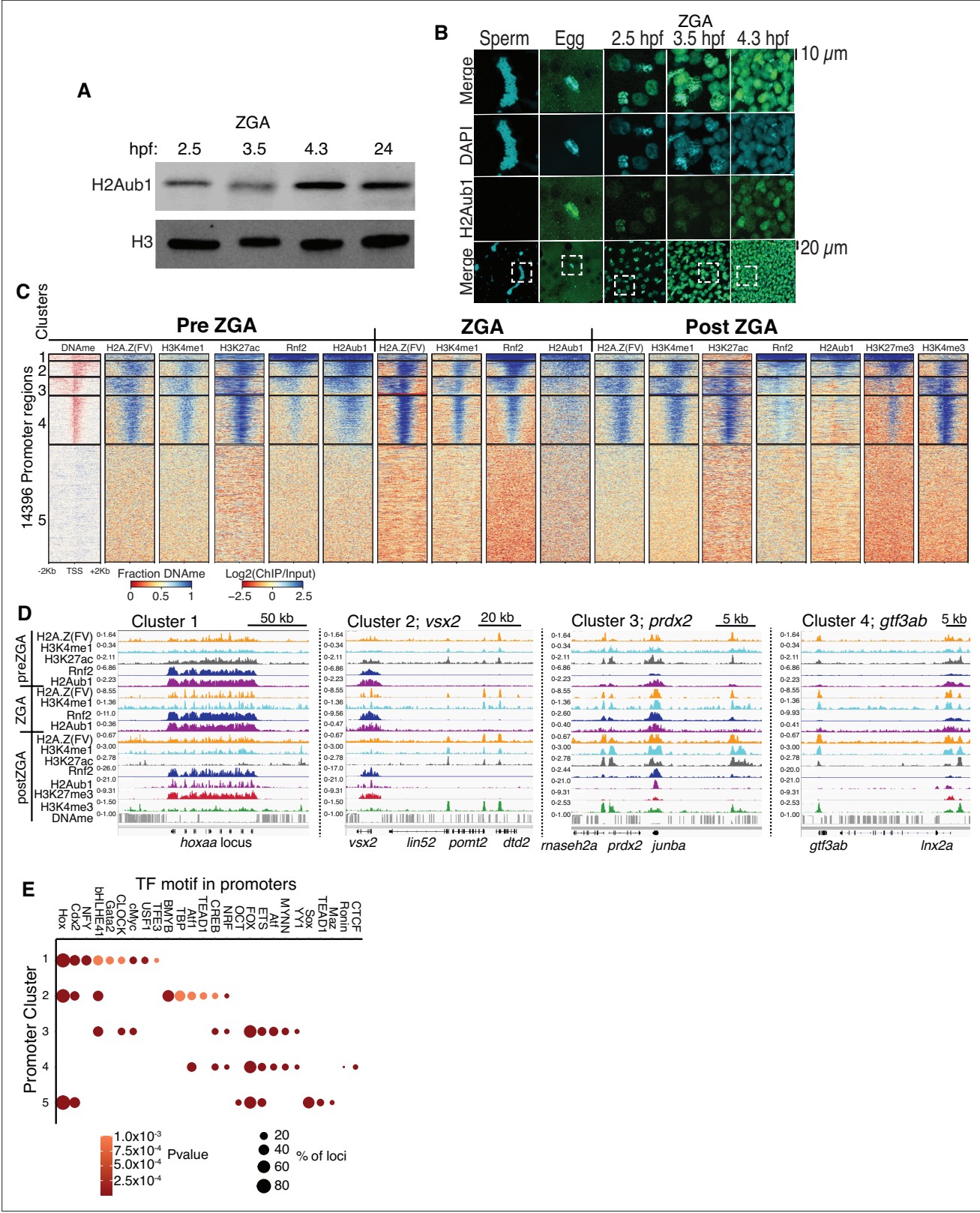

**Figure 1.** Polycomb repressive complex 1 (PRC1) occupancy and activity precedes H3K27me3 establishment at promoters. (**A**) Detection of H2Aub1 and histone H3 (control) by western blot, prior to (2.5 hr post fertilization [hpf]), during (3.5 hpf), following (4.3 hpf) zygotic genome activation (ZGA), and 24 hpf. (**B**) Nuclear H2Aub1 immunofluorescence in zebrafish sperm, oocytes (egg), and embryos prior to (2.5 hpf), during (3.5 hpf), and following (4.3 hpf) ZGA. Dashed square: field of view in upper panels. (**C**) K-means clustering of DNA methylation (DNAme) and chromatin immunoprecipitation (ChIP)-seq

*Figure 1 continued on next page*

*Figure 1 continued*

(histone modifications/variant) at promoters (UCSC refseq). For DNAme, red color indicates regions that lack DNAme. (**D**) Genome browser screenshots of ChIP-seq enrichment at representative genes from the clusters in panel (**C**). (**E**) Transcription factor motif enrichment from HOMER (*Heinz et al., 2010*) at promoter clusters from (**C**).

The online version of this article includes the following source data and figure supplement(s) for figure 1:

**Source data 1.** Uncropped western blots for panel A.

**Figure supplement 1.** Profiling of histone marks in zebrafish embryos and sperm.

**Figure supplement 1—source data 1.** Uncropped western blots for panel C.

**Figure supplement 2.** Chromatin immunoprecipitation (ChIP) replicate structures and additional analyses.

**Figure supplement 3.** Rnf2 and H2Aub1 are not enriched at LINE, LTR, and satellite repetitive elements.

Rnf2 is the sole zebrafish ortholog of Ring1a and Ring1b/Rnf2, the mutually exclusive E3 ligases within mammalian PRC1, which adds monoubiquitin (ub1) to H2A and H2A.Z (*de Napoles et al., 2004*; *Le Faou et al., 2011*; *Wang et al., 2004b*). Rnf2 ChIP-seq at preZGA, ZGA, and postZGA revealed striking coincidence with H2Aub1, and clustering by Rnf2 occupancy revealed five promoter chromatin types, which differed in Rnf2, H2Aub1, and H3K27me3 enrichment, and gene ontology (*Figure 1C*; *Figure 1—figure supplement 2G-K*). Regions with high H2Aub1 and Rnf2 involve broad clustered loci encoding developmental transcription factors (TFs) (cluster 1, e.g. *hoxaa*) or narrow solo/dispersed developmental TFs (cluster 2, e.g. *vsx2*) (*Figure 1C and D*; GO analysis for *Figure 1—figure supplement 2K*). In counter distinction, loci bearing Placeholder and low-moderate levels of H2Aub1 and low Rnf2 largely constitute housekeeping/metabolic genes, with either broad (cluster 3, e.g. *prdx2*) or narrow (cluster 4, e.g. *gtf3ab*) H3K4me3 and H3K27ac occupancy at postZGA (*Figure 1C and D*; *Figure 1—figure supplement 2K*). Notably, 'minor wave' ZGA genes (genes transcribed at 2.5 hr), including those for pluripotency (e.g. *nanog, pou5f3/oct4*), bear marking similar to housekeeping genes, and the robustly transcribed *mir430* locus appears markedly enriched in H3K27ac at preZGA (*Figure 1—figure supplement 2M*; *Chan et al., 2019*). Finally, cluster 5 promoters contain DNAme, and lack Placeholder, H2Aub1, and Rnf2. Thus, over the course of ZGA, loci with Placeholder nucleosomes resolve into two broad classes of loci: developmental genes with high PRC1 and H2Aub1 (clusters 1 and 2), and housekeeping (or 'minor wave') genes that lack substantial PRC1 and H2Aub1, but contain high H3K4me3 and H3K27ac at postZGA (clusters 3 and 4) (*Figure 1C and D*; *Figure 1—figure supplement 2J, K*).

## H3K27me3 establishment occurs during postZGA, and only at locations pre-marked with high Rnf2 and H2Aub1

We find robust H3K27me3 deposition occurring during postZGA at promoters marked during preZGA with high Rnf2 and H2Aub1, specifically at clusters 1 and 2 (*Figure 1C and D*). Furthermore, as embryos transition from preZGA to postZGA, H3K27ac diminishes at developmental loci, whereas housekeeping genes (clusters 3 and 4 *Figure 1C and D*) retain strong H3K27ac and become active. During postZGA, developmental genes acquire the combination of low-moderate H3K4me3 and high H3K27me3, termed 'bivalency' (*Azuara et al., 2006*; *Bernstein et al., 2006*). Interestingly, promoters that become bivalent postZGA involve those pre-marked with higher relative Rnf2 and H2Aub1, whereas promoters with high H3K27ac, high H3K4me3, and low-absent H3K27me3 postZGA involve those pre-marked with lower relative Rnf2 and H2Aub1 (*Figure 1C and D*; *Figure 1—figure supplement 2J*). This observation raised the possibility that high H2Aub1 levels may help recruit PRC2 to subsequently deposit H3K27me3 at developmental genes. Notably, analysis of Rnf2, H2Aub1, and H3K27me3 at LINE, LTR, and satellite repeats revealed no ChIP-seq enrichment at these genomic elements during preZGA, ZGA, or postZGA (*Figure 1—figure supplement 3A-C*), reinforcing that this modification axis is focused on the marking of developmental loci.

To identify candidate TFs that might bind selectively at the promoters of particular clusters, we analyzed the DNA sequences flanking the transcription start site (TSS) (500 bp) at each cluster using the motif finding program, HOMER (*Figure 1E*; *Heinz et al., 2010*). Largely non-overlapping motifs were identified for TF-binding sites at clusters linked to developmental vs. housekeeping genes (*Figure 1E*, clusters 1 and 2 vs. 3 and 4; partitioned by H2Aub1/Rnf2 levels). Here, the strong enrichment of motifs for homeodomain-containing TFs (and other families) in clusters 1 and 2 provides

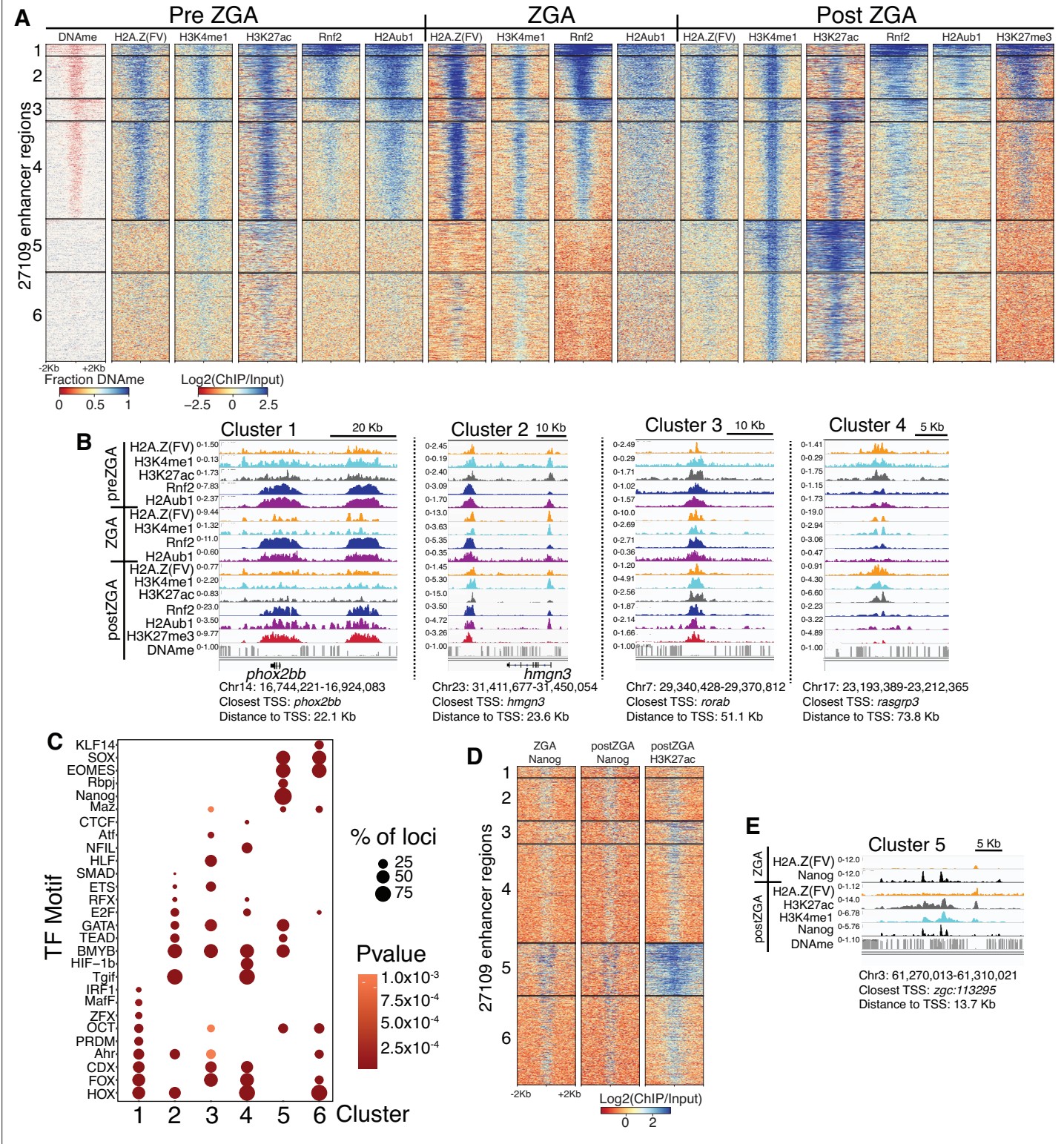

**Figure 2.** Polycomb repressive complex 1 (PRC1) occupancy and activity precedes H3K27me3 at enhancers. (**A**) K-means clustering of whole genome bisulfite sequencing (WGBS) (for DNAme) and chromatin immunoprecipitation (ChIP)-seq at enhancers (postzygotic genome activation [postZGA] H3K4me1 peak summits located outside of promoters). DNAme heatmap displays WGBS fraction-methylated scores (note: red color indicates regions that lack DNAme). ChIP-seq heatmaps display log2(ChIP/input) scores. (**B**) Genome browser screenshots of ChIP-seq enrichment at representative loci from the indicated K-means clusters in (**A**). (**C**) Transcription factor motif enrichment from HOMER (*Heinz et al., 2010*) at enhancer clusters from (**A**). (**D**)

*Figure 2 continued on next page*

*Figure 2 continued*

Features of an enhancer cluster with exceptionally high H3K27ac and Nanog binding. K-means clusters generated in (**A**) were utilized to plot heatmaps of Nanog and H3K27ac. (**E**) A genome browser screenshot depicting Nanog and H3K27ac ChIP enrichment at a DNA-methylated enhancer from cluster 5 in (**D**).

The online version of this article includes the following figure supplement(s) for figure 2:

**Figure supplement 1.** Additional examples of chromatin immunoprecipitation (ChIP) enrichment at enhancers.

candidate factors that may help recruit PRC1 to developmental loci. DNA-methylated promoters at ZGA (cluster 5) represent a large and heterogeneous set of genes, which are activated in particular cell types later in development.

## Enhancer poising parallels features and factors at promoters

Analysis of enhancer regions revealed features that were similar to those at developmental promoters (*Figure 2*). Specifically, enhancers with high H2Aub1 and Rnf2 during preZGA and ZGA acquired robust H3K27me3 during postZGA (*Figure 2A and B*; *Figure 2—figure supplement 1A*). Enhancers with low/absent Rnf2 and H2Aub1 failed to attract robust H3K27me3 at postZGA, instead bearing high levels of H3K27 acetylation (*Figure 2A and B*; *Figure 2—figure supplement 1A*). Candidate TF-binding sites were enriched at enhancers (*Figure 2C*), and these sites overlapped partly with those enriched at promoters, consistent with the expectation that the factors that recruit histone modifiers to promoters and enhancers partially overlap. Taken together, enhancers acquire H3K27me3 during postZGA in proportion to their levels of Rnf2 and H2Aub1 during preZGA and ZGA, consistent with our observations at promoters.

## A unique enhancer class with high H3K27ac and DNA methylation

Curiously, enhancer cluster 5 (*Figure 2A*) was unique at postZGA – displaying high H3K4me1, very high H3K27ac, and open chromatin (via ATAC-seq analysis; *Figure 2—figure supplement 1C,D*) – but bore DNA methylation – an unusual combination given the typical strong correlation between high H3K4me1 and DNA hypomethylation. Notably, Nanog-binding sites were highly enriched solely at cluster 5, and Nanog ChIP-seq during ZGA and postZGA (*Xu et al., 2012*) showed Nanog occupancy highly and selectively enriched at cluster 5 relative to other enhancer clusters (*Figure 2C–E*). Thus, cluster 5 enhancers may utilize Nanog and H3K27ac to open and poise these DNA-methylated enhancers for later/subsequent transition to an active state. Consistent with this notion, GO analysis of cluster 5 enriches for terms related to developmental and signaling processes (p-value; 5.1E-18) (*Figure 2—figure supplement 1B*).

## The PRC2 component Aebp2 is coincident with Rnf2 and H2Aub1 at developmental loci

The pre-marking of developmental genes with H2Aub1 and Rnf2-PRC1 prior to H3K27me3 establishment raised the possibility of a 'non-canonical' (nc) mode of recruitment, involving ncPRC1 action (H2Aub1 addition) followed by the recruitment of PRC2 (ncPRC2) – via H2A/Zub1 recognition – to deposit H3K27me3. This mode and order of recruitment has precedent in *Drosophila* and in mammalian embryonic stem (ES) cell cultures, with the Aebp2 and Jarid2 protein components of ncPRC2 recognizing H2Aub1 and both targeting and facilitating H3K27me3 addition (*Blackledge et al., 2014*; *Blackledge et al., 2020*; *Cooper et al., 2014*; *Cooper et al., 2016*; *Kalb et al., 2014*; *Kasinath et al., 2021*; *Tamburri et al., 2020*). We then addressed whether establishment of H3K27me3 during postZGA is mediated by the non-canonical Aebp2-Jarid2-PRC2 complex at loci pre-marked with H2Aub1. Interestingly, Aebp2 protein levels were very low during preZGA and ZGA stages, but robustly detected postZGA (*Figure 3A*), without a large increase in *aebp2* transcript levels (*Figure 3—figure supplement 1A*). Aebp2 ChIP-seq during postZGA revealed a remarkably high coincidence of Aebp2 with H2Aub1, Rnf2, and H3K27me3 at promoters (*Figure 3B and C*; *Figure 3—figure supplement 1B-D*) and at enhancers (*Figure 3D and E*, *Figure 3—figure supplement 1E*). Taken together, these results suggest that translational upregulation and/or protein stability enables Aebp2 protein accumulation postZGA – enabling the 'reading/binding' of H2Aub1, and H3K27me3 deposition during postZGA by PRC2.

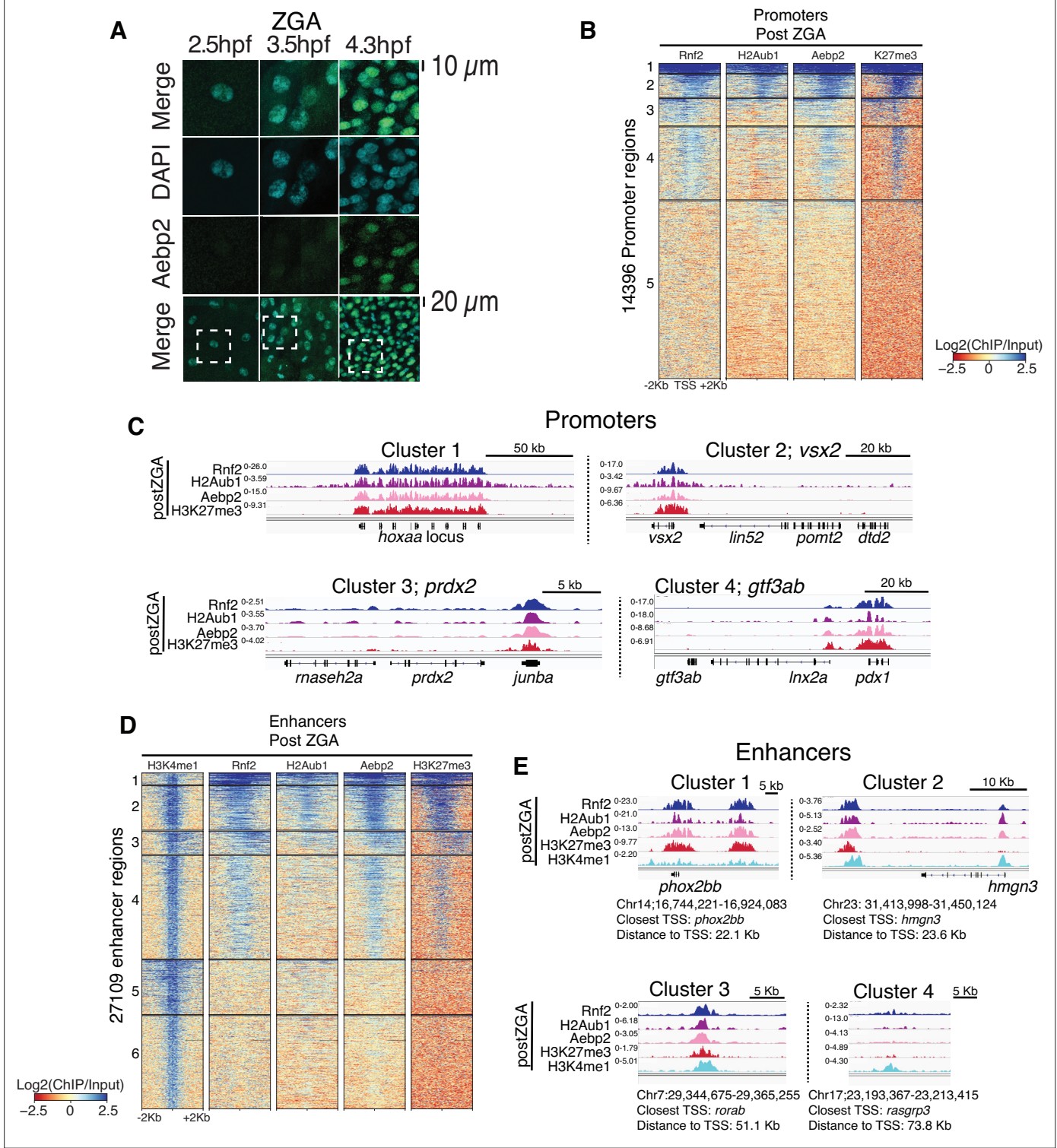

**Figure 3.** Aebp2-polycomb repressive complex 2 (PRC2) mediates de novo H3K27me3 at loci pre-marked by H2Aub1. (**A**) Nuclear Aebp2 detection by immunofluorescence in postzygotic genome activation (postZGA) zebrafish embryos (4.3 hr post fertilization [hpf]). No Aebp2 staining was detected at preZGA (2.5 hpf) or ZGA (3.5 hpf). Bottom row: The dashed square indicates the field of view in upper panels. One of three biological replicates is shown. (**B**) Aebp2 binding at promoters during postZGA overlaps and scales with occupancy of Rnf2, H2Aub1, and H3K27me3. Promoter clusters from *Figure 1C* were utilized to plot heatmaps. (**C**) Genome browser screenshots of chromatin immunoprecipitation (ChIP)-seq at representative promoter loci from clusters in (**B**). (**D**) Aebp2 binding at enhancers during postZGA overlaps with occupancy of Rnf2, H2Aub1, and H3K27me3. Enhancer clusters from *Figure 2A* were utilized to plot heatmaps. (**E**) Genome browser screenshots of ChIP-seq enrichment at representative enhancer loci from clusters

*Figure 3 continued on next page*

*Figure 3 continued*

in (**D**).

The online version of this article includes the following figure supplement(s) for figure 3:

**Figure supplement 1.** Chromatin immunoprecipitation (ChIP) profiling of Aebp2 in postzygotic genome activation (postZGA) embryos.

**Figure supplement 2.** Postzygotic genome activation (postZGA) chromatin immunoprecipitation (ChIP)-seq of Jarid2.

To investigate the possible additional contribution of Jarid2 to 'reading/binding' of H2Aub1 (leading to H3K27me3 deposition), we performed ChIP-seq of Jarid2 during postZGA (***Figure 3—figure supplement 2A***). Here, profiling of Jarid2 occupancy by ChIP-seq resulted in chromatin maps with only modest enrichment and dynamic range, providing 448 bound promoters (***Figure 3—figure supplement 2B***). Comparison of Aebp2 and Jarid2 occupancy at promoters revealed that the majority of Jarid2-binding sites (295/448) overlap with Aebp2-bound sites (***Figure 3—figure supplement 2B***). Only a minority of Aebp2-bound promoters overlapped with Jarid2 – an observation that may reflect our modest ChIP efficiency, but could also reflect the presence of Aebp2-bound promoters that lack Jarid2 binding. Notably, promoters bound by both Aebp2 and Jarid2 had enrichment of GO-term categories corresponding to developmental genes (***Figure 3—figure supplement 2C, D***). Conversely, promoters bound solely by Jarid2 were not associated with developmental functions and instead enriched for ribosomal genes (***Figure 3—figure supplement 2C***). Analysis of Aebp2 and Jarid2 occupancy at enhancer loci revealed similarities to our analysis at promoters. Here, the majority of Jarid2-bound enhancers (84/163) were also bound by Aebp2 (***Figure 3—figure supplement 2E, F***). However, this overlap accounted for only a minority of the Aebp2-bound enhancers. As our Aebp2 ChIP-seq exhibited greater robustness and dynamic range than Jarid2, we hereafter utilized Aebp2 occupancy as the primary functional marker for non-canonical PRC2 complex in the remainder of our analyses.

## Loss of H2Aub1 via Rnf2 inhibition prevents Aebp2 localization and H3K27me3 deposition

To functionally test whether H2Aub1 recruits Aebp2-PRC2 for de novo establishment of H3K27me3 at developmental genes, we utilized the RNF2 inhibitor, PRT4165 (***Chagraoui et al., 2018***; ***Ismail et al., 2013***; ***Zhu et al., 2018***). PRT4165 is a small molecule inhibitor of Rnf2 that has previously been shown to strongly reduce H2Aub1 modification, but not to affect the activity of related H2A E3 ligases such as Rnf8 and Rnf168 (***Chagraoui et al., 2018***; ***Ismail et al., 2013***; ***Zhu et al., 2018***). In each of three biological replicates, PRT4165 treatment (150 µM) from the one-cell stage onward largely eliminated H2Aub1 by 4 hpf (ZGA) (***Figure 4A and B***), and conferred a developmental arrest that resembled untreated 4 hpf embryos (***Figure 4—figure supplement 1A***). To determine whether the loss of H2Aub1 conferred loss of Aebp2 genomic targeting, we performed ChIP experiments on Aebp2 in PRT4165-treated and DMSO-treated embryos (three biological replicates per condition). Remarkably, developmental loci that normally display high Aebp2 in untreated or DMSO-treated embryos lost Aebp2 binding following PRT4165 treatment (***Figure 4C***, clusters 3 and 4; ***Figure 4—figure supplement 1B, C***). Curiously, PRT4165 treatment also conferred many new/ectopic Aebp2 peaks (***Figure 4C***, clusters 1 and 2), however our profiling of H3K27me3 following PRT4165 treatment (three biological replicates) revealed that new/ectopic Aebp2 sites did not acquire H3K27me3 (***Figure 4C***, clusters 1 and 2; ***Figure 4—figure supplement 2B***), consistent with prior observations that other modifications (such as H2Aub1; ***Kalb et al., 2014***) may be needed to stimulate H3K27me3 addition by PRC2. Importantly, treatment with PRT4165 eliminated or strongly reduced H3K27me3 at virtually all loci normally occupied by Aebp2 and H3K27me3 (***Figure 4C***, clusters 3 and 4, ***Figure 4—figure supplement 1D***), a conclusion supported by our use of a 'spike-in' control involving *Drosophila* nuclei bearing H3K27me3-marked regions in all three replicates per condition. Finally, new ectopic H3K27me3 peaks were very rare in the PRT4165 treatment, and none of these loci were bound by Aebp2 (***Figure 4—figure supplement 2B***). Taken together, H2Aub1 deposition by Rnf2 during preZGA is required for the recruitment of Aebp2 and subsequent de novo deposition of H3K27me3 postZGA at developmental loci.

To determine whether H2Aub1 impacts transcriptional repression of developmental genes at postZGA, we performed RNA sequencing (three biological replicates per condition) on 4 hpf embryos that were either vehicle-treated (DMSO) or PRT4165-treated from the one-cell stage onward

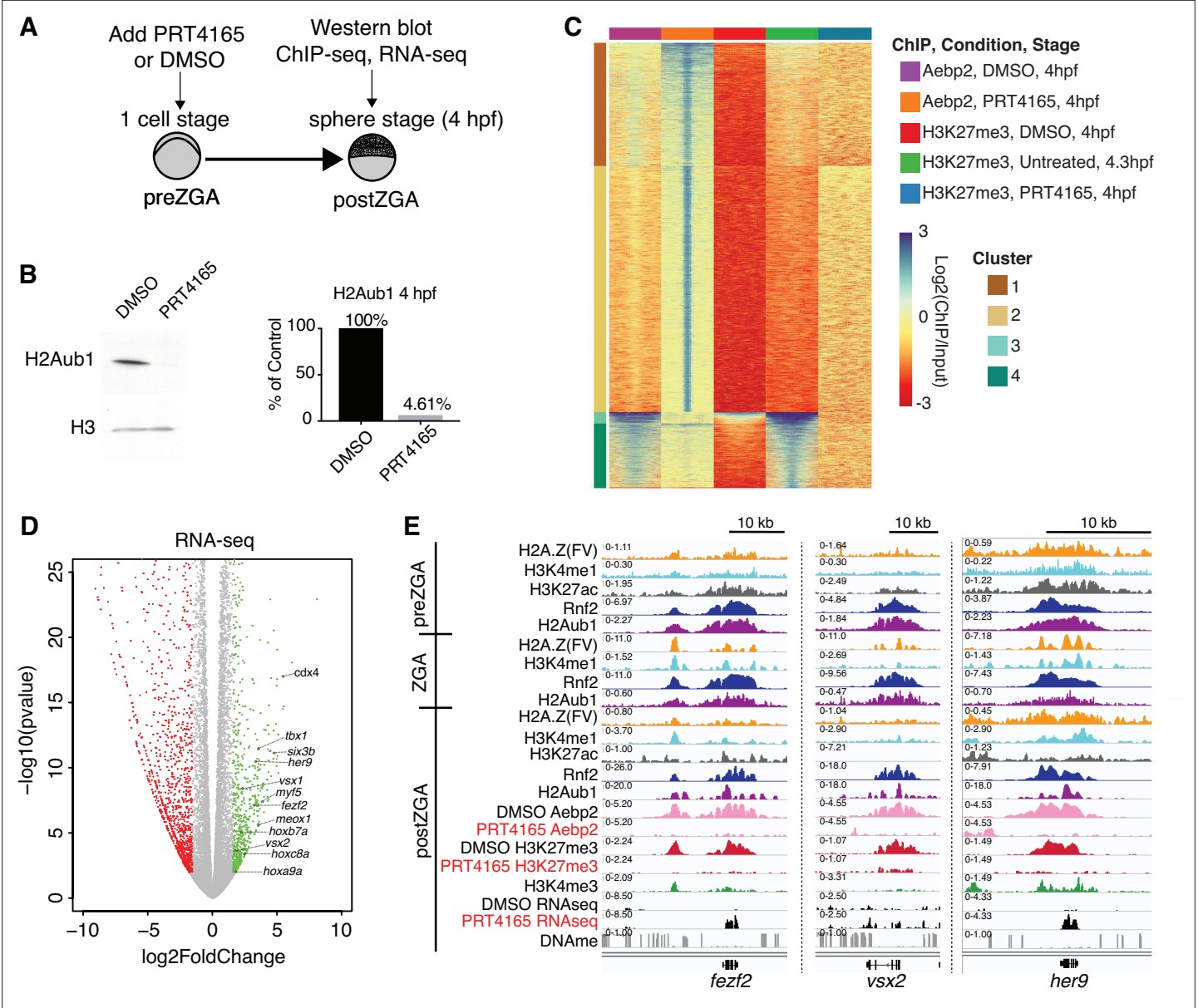

**Figure 4.** Catalytic activity of polycomb repressive complex 1 (PRC1) is required for aebp2 binding, H3K27me3 establishment, and transcriptional repression of developmental genes. (**A**) Experimental design of drug treatments to inhibit Rnf2 activity. Embryos at the one-cell stage were added to media containing either PRT4165 (150 μM) or DMSO and raised until 4 hr post fertilization (hpf). (**B**) PRT4165 treatment of embryos confers bulk loss of H2Aub1 at 4 hpf. Left: Western blot for H2Aub1 in 4 hpf embryos treated with DMSO (vehicle) or 150 μM PRT4165 (Rnf2 inhibitor). Right: Quantification western blot in left panel. (**C**) Impact of Rnf2 inhibition on Aebp2 genomic localization and H3K27me3. K-means clustering of Aebp2 and H3K27me3 chromatin immunoprecipitation (ChIP)-seq enrichment at all loci with called peaks in any of the datasets plotted. Embryos were treated from the one-cell stage with either DMSO or 150 μM PRT4165 and harvested at 4 hpf for ChIP analysis. H3K27me3 ChIP-seq from untreated embryos at 4.3 hpf is plotted as a comparitor. (**D**) Impact of Rnf2 inhibition on gene expression. Volcano plot of RNA-seq data from PRT4165-treated vs. untreated embryos (4 hpf). Green and red data points signify transcripts with p-values < 0.01 and at least a 3-fold change (increase or decrease) in expression, respectively. Marquee upregulated genes encoding developmental transcription factors are labelled. (**E**) Genome browser screenshots of representative developmental genes which, upon Rnf2 inhibition, lose Aebp2 binding and H3K27me3 marking, and become transcriptionally active.

The online version of this article includes the following source data and figure supplement(s) for figure 4:

**Source data 1.** Uncropped western blots for panel B.

**Figure supplement 1.** Replicate structures of chromatin immunoprecipitation (ChIP)-seq and RNA-seq experiments involving drug treatments.

**Figure supplement 2.** Loss of H2Aub1 results in disrupted Aebp2 localization, H3K27me3 marking, and transcriptional de-repression.

**Figure supplement 3.** RNAseq of PRT4165-treated embryos at prezygotic genome activation (preZGA) and ZGA.

(*Figure 4—figure supplement 1E*). Here, we identified and characterized both up- and downregulated genes (*Figure 4E*). PRT4165-upregulated and -downregulated genes (>3-fold, p-value < 0.01) were both enriched in developmental factors, but the number of genes associated with upregulated GO-terms was substantially greater than downregulated GO-terms (*Figure 4—figure supplement 1F*). Here, ~16.6% of H2Aub1-marked protein coding genes were upregulated, which may reflect the availability at postZGA of an opportunistic activator, following H2Aub1 loss (*Figure 4—figure supplement 2C*). Affected genes include those in clustered loci (e.g. *Hox* genes) where the effect was moderate, as well as non-clustered/solo formats where the effect of PRT4165 was more pronounced (*Figure 4E and F*; *Figure 4—figure supplement 2A*).

Having observed precocious developmental gene upregulation in response to PRT4165 treatment during postZGA (4 hpf), we were curious whether their upregulation could also be observed during the preZGA and ZGA stages. To test, we repeated our PRT4165 treatment regimen and isolated embryos at preZGA (2.5 hpf) and ZGA (3.5 hpf) stages, and performed RNAseq (with biological triplicates; *Figure 4—figure supplement 3*). Here, analysis of upregulated genes meeting our threshold criteria (fold change ≥1.5; p-value ≤ 0.01) by GO-term analysis uncovered no enrichment of developmental genes upon RNF2 inhibition during preZGA or ZGA. Instead, enriched GO-terms from upregulated transcripts corresponded to genes encoding RNA-binding proteins and ribosomal proteins. Thus, chromatin de-repression (via H2Aub1 removal) does not cause transcriptional activation of large numbers of developmental genes during preZGA or ZGA – not even the developmental genes that are activated postZGA following Rnf2 inhibition. Here, we note that general/housekeeping transcription does not occur until ZGA; only the miRNA-430 locus and a very limited number of genes are transcribed during preZGA. Therefore, H2Aub1 removal does not, by itself, lead to the activation of developmental genes that are normally marked by H2Aub1 during preZGA. Taken together, these results strongly suggest that H2Aub1 represses developmental genes during preZGA and ZGA, independent of H3K27me3, and that the absence of H2Aub1 renders developmental genes susceptible to precocious activation following ZGA, which confers developmental arrest prior to gastrulation.

## Discussion

Our work reveals that developmental gene silencing in early zebrafish embryos is established through sequential recruitment and activity of PRC1 and PRC2 complexes, respectively, to otherwise open/permissive loci bearing Placeholder nucleosomes (*Figure 5*). Placeholder nucleosomes containing H3K4me1 and the histone variant H2A.Z(FV) are installed by the chromatin remodeler SRCAP during preZGA (*Murphy et al., 2018*), and are focally pruned to small regions by the chaperone Anp32e (*Kobor et al., 2004*; *Krogan et al., 2003*; *Mao et al., 2014*; *Obri et al., 2014*; *Wang et al., 2016*). Here, our reanalysis of published data confirms that H3K27ac is an additional component of Placeholder nucleosomes during preZGA (*Zhang et al., 2018*). Functional studies reveal that Placeholder nucleosomes prevent DNAme where they are installed and are utilized to reprogram the DNAme patterns during cleavage stage. Therefore, from a 'permissive' Placeholder platform during preZGA, two very different chromatin/transcriptional states are attained at ZGA: active or poised (*Murphy et al., 2018*).

Our work suggests that the initial poised state at developmental genes and enhancers involves the imposition of polycomb-based silencing upon the permissive states established by Placeholder nucleosomes. Specifically, we observe H2Aub1 addition by Rnf2/PRC1 during preZGA and ZGA to confer initial transcriptional silencing at developmental loci – which is subsequently read by the Aebp2/PRC2 complex to add H3K27me3 after ZGA (*Figure 5*). Prior work in ES cells has provided an in vitro parallel in which PRC1 activity (H2Aub1 addition) can occur independently of PRC2-mediated recruitment at certain loci (*Blackledge et al., 2014*; *Blackledge et al., 2020*; *Cooper et al., 2014*; *Cooper et al., 2016*; *Kalb et al., 2014*; *Tamburri et al., 2020*; *Tavares et al., 2012*), which has been termed 'non-canonical' order of recruitment, to contrast with prior data showing the reverse/canonical order (*Wang et al., 2004a*). Furthermore, and consistent with our work, human AEBP2 and JARID2 have recently been shown to directly bind H2Aub1 and stimulate PRC2 activity in the presence of H3K4 methylation (*Kasinath et al., 2021*), and a similar mechanism may be utilized to establish bivalency after ZGA in zebrafish.

Our work also clarifies and extends prior work in zebrafish which showed that an incross of zebrafish heterozygous for an *rnf2* loss-of-function truncation mutation yielded a pleitropic terminal phenotype

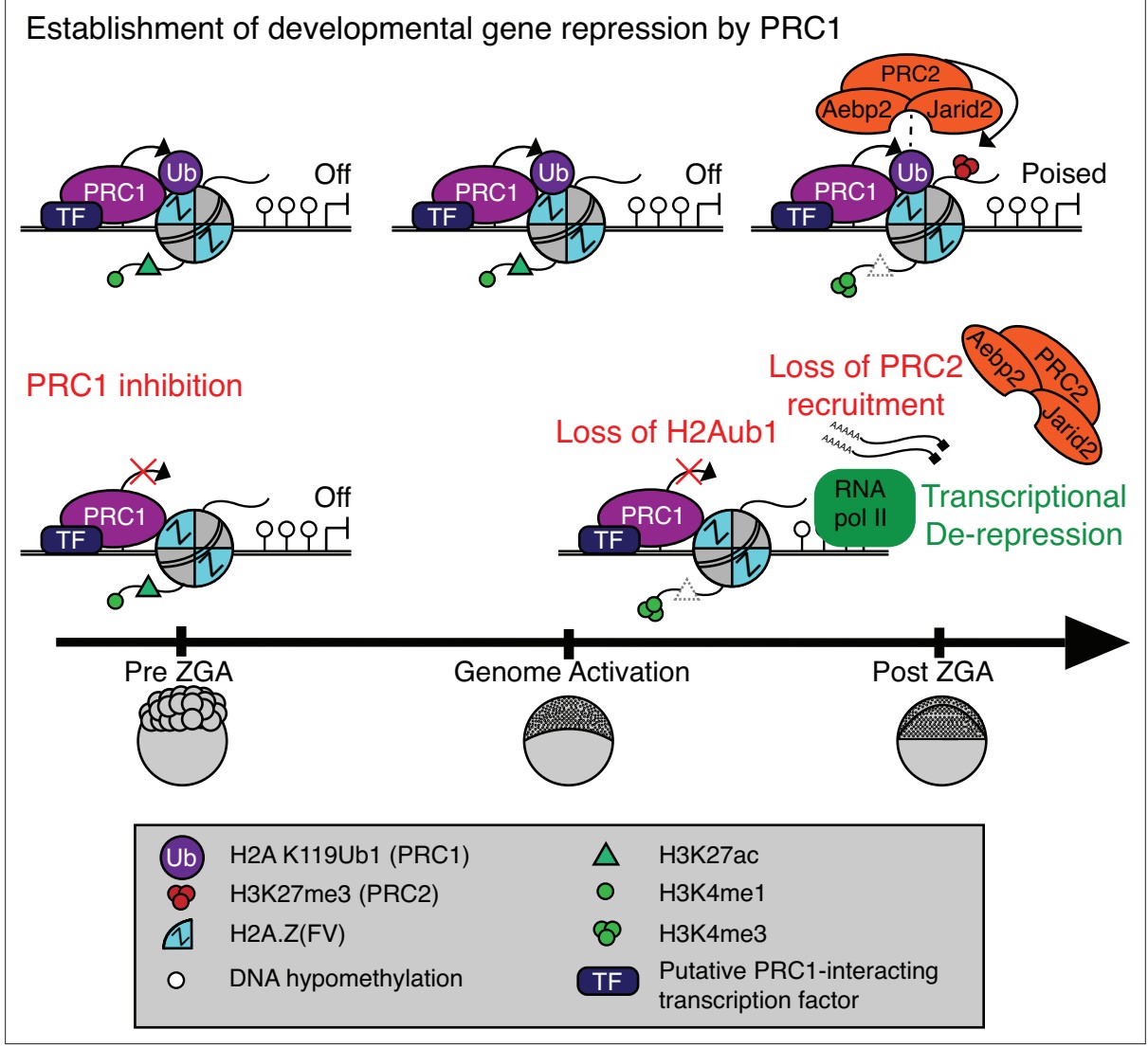

**Figure 5.** Model for polycomb-mediated establishment of developmental gene silencing during zebrafish embryogenesis. Prior to zygotic genome activation (ZGA), Rnf2-PRC1 is recruited by transcription factors (TFs) to promoters (shown) and enhancers (not shown) of developmental genes bearing Placeholder nucleosomes (H2A.Z(FV), H3K4me1, H3K27ac). Rnf2-PRC1 deposition of H2Aub1 recruits Aebp2-PRC2 to catalyze H3K27me3 addition. H2Aub1 ablation (via Rnf2 inhibition) eliminates Aebp2-PRC2 recruitment and prevents H3K27me3 establishment. Notably, H2Aub1 loss causes precocious transcription of certain developmental genes after ZGA, identifying H2Aub1 as a critical component of silencing at ZGA.

at 3 days post fertilization (dpf), coincident with the loss of *rnf2* RNA. Notably, loss of *rnf2* at 3 dpf was associated with partial upregulation of certain developmental genes, but H3K27me3 remained fully present, showing that Rnf2 is not required for H3K27me3 maintenance (*Chrispijn et al., 2019*). However, progression of *rnf2* mutants to 3 dpf may have relied on maternally inherited WT *rnf2* RNA or protein to provide the initial establishment of gene silencing and H3K27me3, raising the possibility that *rnf2* is actually essential at a much earlier developmental stage. Our use of the Rnf2 inhibitor, PRT4165, during preZGA and ZGA stages reveals the necessity for Rnf2 activity for the establishment of developmental gene silencing, for subsequent H3K27me3 addition, and for progression of zebrafish development beyond the ZGA stage. Importantly, developmental gene upregulation is not attributable to H3K27me3 loss, as maternal zygotic *ezh2* mutant zebrafish embryos do not precociously activate developmental genes during ZGA, and they progress through gastrulation without H3K27me3 (*Rougeot et al., 2019*; *San et al., 2016*; *San et al., 2019*). Notably, although upregulation of developmental genes in the presence of PRT4165 is clear, this involves ~16.6% of the developmental gene

repertoire occupied by H2Aub1/Rnf2. Here, we suggest that tissue-specific activators for the majority of developmental genes are not present at ZGA.

Maternal loading of mRNA and proteins present challenges for functional experiments during early embryonic stages of development. Our use of the Rnf2 inhibitor, PRT4165, enabled us to test whether H2Aub1 impacts gene expression at ZGA, and whether H2Aub1 is required for the recruitment of Aebp2-PRC2 and for the subsequent establishment of H3K27me3 at developmental loci during postZGA. The observable impact of PRT4165 includes the loss of H2Aub1, the loss of Aebp2 recruitment to all loci that formerly bore H2Aub1, and the loss of H3K27me3 deposition at all developmental loci that are normally marked by H3K27me3 during postZGA. Our interpretation of these observations is that they are linked to, and dependent on, Rnf2 inhibition. However, it is important to note that PRT4165 may have off-target effects aside from inhibition of Rnf2 which may contribute to our observations. While we cannot formally account for possible off-target effects of PRT4165, we note that ZGA (which involves the activation of thousands of genes) progresses relatively normally, except for the upregulation of a cohort of developmental genes which were formerly H2Aub1 marked. This provides a measure of confidence that the upregulation of developmental genes is largely a direct consequence of H2Aub1 loss in the early embryo. However, as PRT4165 was the only approach we employed that successfully removed the vast majority of H2Aub1, future studies involving orthogonal approaches to elicit Rnf2 loss in zebrafish will be needed to further validate the consequences of H2Aub1 loss in early zebrafish embryos.

While our work was in review, work on H2Aub1 and H3K27me3 dynamics in the preimplantation mouse embryo was reported by two separate groups (*Chen et al., 2021*; *Mei et al., 2021*). Both studies in mouse demonstrated that H2Aub1 temporally precedes H3K27me3 at developmental genes during early embryonic stages, and both studies also lowered H2Aub1 via one effective approach: either through maternal genetic loss of non-canonical PRC1 subunits (*Pcgf1/6*) or through overexpression of an H2A de-ubiquitinase (PR-DUB). Here, both approaches led to precocious transcription of a moderate subset of developmental genes at or shortly after ZGA. Thus, these sets of results align well with our observations in zebrafish.

Notably, each method of H2Aub1 perturbation in mice was unique (with attendant advantages and caveats) and conferred different impacts on embryogenesis and establishment of H3K27me3 at developmental loci. In mice, zygotic overexpression of PR-DUB resulted in rapid and significant erasure of H2Aub1, precocious transcription of certain developmental genes, and growth arrest at the four-cell stage. However, reduction of H2Aub1 by zygotic PR-DUB overexpression had minimal impact on H3K27me3, likely due to the fact that H3K27me3 is very low at the four-cell stage. In contrast, Mei et al. disrupted H2Aub1 via maternal genetic ablation (paternal genes remained intact) of two genes, *Pcgf1/6*, which encode subunits of noncanonical PRC1. Here, *Pcgf1/6* mutant oocytes were fertilized with wild-type sperm to produce maternal-null *Pcgf1/6* embryos. In contrast to PR-DUB overexpression, maternal-null *Pcgf1/6* embryos displayed global reductions in both H2Aub1 and H3K27me3 during cleavage stages – a finding that aligns well with our findings. Notably, the reductions of H2Aub1 and H3K27me3 from maternal *Pcgf1/6* loss largely returned to wild-type levels by the morula stage, possibly due to zygotically (and paternally) expressed *Pcgf1/6*. Here, we postulate that the more severe loss of H3K27me3 from *Pcgf1/6* deletion may stem from the temporal difference of disrupting H2Aub1 in the female germline vs. the early mouse embryo, or potentially the developmental stage in which H3K27me3 was assayed (two cell vs. four cell). A major issue not addressed in these recent papers in mouse is how mechanistically H2Aub1 directs H3K27me3. Notably, our study addresses this mechanism by demonstrating that non-canonical PRC2 complex containing the H2Aub1 'reader' Aebp2 occupies all loci with high H2Aub1 during postZGA, which then receive H3K27me3.

Interestingly, Chen et al. and Mei et al. utilized hybrid mouse strains to trace parental asymmetries of H3K27me3 and H2Aub1 in the early embryo. Here, both groups uncovered rapid erasure of H2Aub1 from the paternal genome shortly after fertilization which is re-established by the two-cell stage. Our data also demonstrate that the paternal genome is initially devoid of H2Aub1, and rapidly acquires H2Aub1 shortly after fertilization. However, zebrafish sperm are entirely devoid of H2Aub1 (*Figure 1B*; *Figure 1—figure supplement 1C*), whereas mouse sperm bear H2Aub1 coincident with H3K27me3. Thus, the paternal genome in both mammals and zebrafish attain a transient state lacking H2Aub1 soon after fertilization – but arrive at that state by alternative routes.

Here, we observe that developmental or housekeeping gene promoters attract either a high- or low-level PRC1 binding, respectively. How ncPRC1 is recruited to CpG islands in ES cells is partially understood, as the ncPRC1 subunit Kdm2b helps recruit ncPRC1 to CpG islands of developmental genes. In this context, Kdm2b binds to hypomethylated CpGs via the CxxC motif (*Blackledge et al., 2014*; *Farcas et al., 2012*; *He et al., 2013*; *Wu et al., 2013*). However, ncPRC1 is recruited robustly to only a minority of Kdm2b-bound CpG islands, implying that additional factors are needed to specify recruitment of ncPRC1 to CpG islands of developmental genes. Here, we speculate that particular TFs, such as the candidates in *Figures 1E and 2C*, are likewise utilized in cooperation with zebrafish Kdm2b to enable strong focal recruitment of Rnf2-PRC1 to Placeholder-occupied developmental loci. In contrast, the candidate TFs at housekeeping genes would recruit MLL complexes to implement H3K4 methylation, and not Rnf2-PRC1. Thus, future work will explore which particular TFs recruit SRCAP (for Placeholder/H2A.Z installation), PRC1 (for H2Aub1 addition at developmental genes), or MLL complexes (for H3K4me3 modification at housekeeping genes) to define which specific genes are subjected to silencing or activation at ZGA.

One curiosity arising from our work is why zebrafish utilize the non-canonical PRC1 complex, which adds monoubiquitination to H2A/H2A.Z(FV), rather than canonical PRC1 complex, which compacts chromatin for initial developmental gene silencing. Here, we speculate that the rapid (~16 min) cell cycles that characterize the preZGA cleavage state – coupled to the need for continual DNA replication during cleavage stage – are not compatible with the compaction conferred by canonical PRC1 (*Gao et al., 2012*; *Grau et al., 2011*; *Lau et al., 2017*). Furthermore, the necessary substrate to recruit canonical PRC1 to chromatin, H3K27me3, is absent in preZGA embryos. Instead, the use of the repression modes conducted by non-canonical PRC1 addition of H2Aub1 – which antagonizes RNA Pol II transcriptional initiation or bursting – may help confer silencing without conferring a compaction that might impede DNA replication (*Dobrinić et al., 2020*; *Stock et al., 2007*). However, once embryos exit cleavage stage, the cell cycle greatly lengthens, and Aebp2-PRC2 complexes add H3K27me3 to loci – which may then enable canonical PRC1 to localize to H3K27me3-marked loci, and conduct compaction. Indeed, prior work has shown that later stages of zebrafish development utilize canonical PRC complexes (*Rougeot et al., 2019*). Here, future studies may reveal more precisely the logic and timing underlying the transition from non-canonical to canonical utilization of PRC complexes in zebrafish development.

# Materials and methods

## Key resources table

| Reagent type (species) or resource | Designation | Source or reference | Identifiers | Additional information |
|---|---|---|---|---|
| Gene (*Danio rerio*) | Zebrafish Genome | UCSC | Zv10 | |
| Antibody | Anti-H2Aub1 (rabbit monoclonal) | Cell Signaling Technology | Cat# 8240; RRID:AB_10891618 | WB (1:1000) ChIP (1:100) IF (1:500) |
| Antibody | Anti-rnf2 (rabbit monoclonal) | Cell Signaling Technology | Cat# 5694; RRID:AB_10705604 | (1:100) |
| Antibody | Anti-H2A.Z (rabbit polyclonal) | Active Motif | Cat# 39113; RRID: AB_2615081 | (5 µl) |
| Antibody | Anti-H3K4me1 (rabbit polyclonal) | Active Motif | Cat# 39297; RRID: AB_2615075 | (10 µl) |
| Antibody | Anti-aebp2 (rabbit monoclonal) | Cell Signaling Technology | Cat# 14,129 S; RRID: AB_2798398 | ChIP (1:100) IF (1:500) |
| Antibody | Anti-H3K27me3 (rabbit polyclonal) | Active Motif | Cat# 39155; RRID: AB_2561020 | ChIP (5 µl) IF (1:500) |
| Antibody | Anti-H3 (mouse monoclonal) | Active Motif | Cat# 39763; RRID: AB_2650522 | (1:2000) |
| Antibody | Anti-Jarid2 rabbit polyclonal | Novus | Cat# NB100-2214; RRID:AB10000529 | (5 µl) |

*Continued on next page*

*Continued*

| Reagent type (species) or resource | Designation | Source or reference | Identifiers | Additional information |
|---|---|---|---|---|
| Cell line (*Danio melanogaster*) | Cell line S2: S2-DRSC | ATCC | Cat# CRL-1963 RRID:CVCL_Z232 | |
| Chemical compound, drug | PRT4165 | Tocris | Cat# 5047 | |
| Commercial assay or kit | NEBNext ChIP-Seq Library Prep Master Mix Set for Illumina | New England Biolabs | Cat# E6240 | . |
| Commercial assay or kit | Illumina TruSeq Stranded mRNA Library Prep Kit | Illumina | Cat# RS-122–2101, RS-122–2102 | |
| Software, algorithm | Novoalign | Novocraft | RRID:SCR_014818 | |
| Software, algorithm | Samtools | *Li et al., 2009* | RRID:SCR_002105 | |
| Software, algorithm | deepTools | *Ramírez et al., 2016* | RRID:SCR_016366 | |
| Software, algorithm | MACS2 | *Zhang et al., 2008* | RRID:SCR_013291 | |
| Software, algorithm | UCSC Exe Utilities | UCSC Genome Browser | http://hgdownload.soe.ucsc.edu/downloads.html#source_downloads | |
| Software, algorithm | IGV | *Thorvaldsdóttir et al., 2012* | RRID:SCR_011793 | |
| Software, algorithm | DAVID | *Huang et al., 2009* | https://david.ncifcrf.gov | |
| Software, algorithm | R | *R Development Core Team, 2020* | RRID:SCR_001905 | |
| Software, algorithm | R Studio | *RStudio Team, 2021* | RRID:SCR_000432 | |
| Software, algorithm | Chipseekr | *Yu et al., 2015* | https://www.bioconductor.org/packages/release/bioc/html/ChIPseeker.html | |
| Software, algorithm | Bio-ToolBox | Tim Parnell of the Huntsman Cancer Institute | *Parnell, 2021b*; https://github.com/tjparnell/biotoolbox | |
| Software, algorithm | Multi-Replica Macs ChIPSeq Wrapper | Tim Parnell of the Huntsman Cancer Institute | *Parnell, 2021a*; https://github.com/HuntsmanCancerInstitute/MultiRepMacsChIPSeq | |
| Software, algorithm | HOMER | *Heinz et al., 2010* | RRID:SCR_010881 | |
| Software, algorithm | STAR | *Dobin et al., 2013* | RRID:SCR_015899 | |
| Software, algorithm | featureCounts | *Liao et al., 2013* | RRID:SCR_012919 | |
| Software, algorithm | DESeq | *Anders and Huber, 2010* | RRID:SCR_000154 | |

## Zebrafish husbandry

Wild-type Tübingen zebrafish were maintained as described (*Westerfield, 2007*). All experiments involving zebrafish were approved by University of Utah IACUC (Protocol 20–04011). Embryos were scored for developmental staged as described (*Kimmel et al., 1995*).

## Acid extraction of nuclear proteins

Two-hundred embryos were added to 1.5 ml tubes and washed twice with cold PBS; 800 µl of Mild Cell Lysis Buffer (10 mM Tris-HCl pH 8.1, 10 mM NaCl, 0.5 NP-40, 2× protease inhibitors) was applied to embryos and incubated on ice for 5 min. Embryos were homogenized by passing through a 20-gauge syringe several times. Samples were briefly centrifuged to bring down chorions. Supernatants were transferred to fresh tubes and centrifuged at 1300× *g* for 5 min at 4°C. Pelleted nuclei were washed twice with cold Mild Cell Lysis Buffer. Nuclei were resuspended to a final volume of 800 µl in cold Mild Cell Lysis Buffer and supplemented with 10 µl of sulfuric acid (18.4 M). Samples were sonicated for 10 s (1 s ON, 0.9 s OFF) at 30% output using a Branson sonicator. Proteins were extracted for 30 min at 4°C on a rotator, and 160 µl of 100% trichloroacetic acid was added and proteins were allowed to precipitate for 30 min on ice. Samples were centrifuged at 13,000 rpm for 5 min at 4°C. Protein pellets were washed with 800 µl of cold acidified acetone, and centrifuged again at 13,000 rpm for 5 min at 4°C. Protein pellets were washed with 800 µl of cold acetone, and centrifuged again at 13,000 rpm

for 5 min at 4°C. Supernatant was discarded and pellets were dried at 37°C for 5 min. Dried protein pellets were resuspended in 2× Laemmli sample buffer and boiled for 8 min. Samples were then used for western blotting. Bands were quantified in ImageJ (*Schneider et al., 2012*).

## Immunohistochemistry and DAPI staining

Standard protocol for immunohistochemistry was followed as described (*Fernández and Fuentes, 2013*; *Zhang et al., 2018*). Three biological replicates were performed for each immunohistochemistry experiment. Briefly, 30 embryos were collected at appropriate timepoints and fixed with fresh 4% paraformaldehyde (Electron Microscopy, Cat# 50980487) in 1× PBS at room temperature for 12 hr. Droplets of glacial acetic (100%, Merck, Cat# 1000560001) or DMSO (final concentration 0.5%, Sigma) were added 5–10 s after initiation of the fixation. Chorions were manually removed from fixed embryos with forceps and dechorionated embryos were dehydrated in methanol and stored at –20°C. For immune-staining, embryos were rehydrated into PB3T (1× PBS with 0.3% TritonX-100, and then incubated in blocking agent 1% BSA, 0.3 M glycine in PB3T). Embryos were incubated with primary antibodies diluted in blocking agent overnight at 4°C. Primary antibodies were removed and embryos were washed extensively with PB3T. Embryos were next incubated with appropriate secondary antibodies in the dark followed by extensive washes in PB3T. Primary antibodies used for immune-staining are listed below. Secondary antibodies used were donkey α-rabbit IgG-488 at 1:500 (Life Technologies, Cat# A-21206). DAPI was used at 1:1000 as a nuclear counterstain. The yolk cells were removed from embryo and embryo was mounted on glass slide with ProLong Gold Antifade mounting media (Thermo Fisher, Cat# P-36931) and a 2.0 mm square coverslip sealed with nail polish. Samples were stored at 4°C until imaged.

## Imagining of zebrafish embryos

Images were acquired on a Leica SP8 White Light laser confocal microscope. Image processing was completed using Nikon NIS-Elements multi-platform acquisition software with a 40×/1.10 Water objective. Fiji (ImageJ, V 2.0.0-rc-69/1.52p) was utilized to color DAPI channel to cyan, GFP color remained green. Confocal images are max projections of Z stacks taken 0.5 μm apart for a total of the embryo ~7–12 μm.

## Primary antibodies

The following antibodies were utilized in for the present study: anti-H2Aub1 (Cell Signaling Technology Cat# 8240; RRID:AB_10891618), anti-Rnf2 (Cell Signaling Technology Cat# 5694; RRID:AB_10705604), anti-H2A.Z (Active Motif Cat# 39113; RRID:AB_2615081), anti-H3K4me1 (Active Motif Cat# 39297; RRID:AB_2615075), anti-Aebp2 (Cell Signaling Technology Cat# 14129; RRID: AB_2798398), anti-H3K27me3 (Active Motif Cat# 39155; RRID: AB_2561020), anti-H3 (Active Motif Cat# 39763; RRID:AB_2650522), and anti-Jarid2 (Novus Cat# NB100-2214; RRID:AB10000529).

## ChIP-seq in zebrafish embryos

### Embryo fixation

Approximately 1.5 million cells were used for each ChIP replicate. Embryos were allowed to progress to the desired developmental stage and then transferred to 1.5 ml microcentrifuge tubes (~200 embryos per tube). Chorions were removed enzymatically by treatment with pronase (1.25 mg/ml in PBS). Dechorionated embryos were gently washed twice with PBS to remove pronase. Samples were fixed with 1% formaldehyde (Electron Microscopy Sciences, Cat# 15712) for 10 min at room temperature with end over end rotation. Fixation was quenched with 130 mM glycine for 5 min at room temperature. Samples were centrifuged for 5 min at 500× *g* at 4°C. Supernatant was discarded and cell pellets were washed twice with ice-cold PBS. Cell pellets were frozen with liquid nitrogen and stored at –80°C.

### Nuclei isolation and lysis

One ml of Mild Cell Lysis Buffer (10 mM Tris-HCl pH 8.1, 10 mM NaCl, 0.5 NP-40, 2× proteinase inhibitors) was applied to cell pellets from 1000 embryos and rotated at 4°C for 10 min. Samples were centrifuged at 1300× *g* for 5 min at 4°C. Supernatant was discarded and nuclei pellets were resuspended in 1 ml Nuclei Wash Buffer (50 mM Tris-HCl pH 8.0, 100 mM NaCl, 10 mM EDTA, 1% SDS, 2×

protease inhibitors) and rotated at room temperature for 10 min. Samples were centrifuged at 1300× $g$ for 5 min at 4°C to pellet nuclei. Supernatant was discarded and nuclei pellets were resuspended in 100 µl of Nuclei Lysis Buffer (50 mM Tris-HCl pH 8.0, 10 mM EDTA, 1% SDS, 2× proteinase inhibitors). Samples were incubated on ice for 10 min; 900 µl of IP Dilution Buffer (16.7 mM Tris-HCl pH 8.1, 167 mM NaCl, 1.2 mM EDTA, 0.01% SDS, 1.1% Triton X-100, 2× proteinase inhibitors) was added to samples.

### Chromatin sonication
Nuclear lysates were sonicated with a Branson Digital Sonifier with the following settings: 10 s duration (0.9 s ON, 0.1 s OFF), 30% amplitude. Seven sonication cycles were performed. Samples were placed in an ice bath for at least 1 min between each sonication cycle. Sonicated samples were centrifuged at 14,000 rpm, 4°C, for 10 min to pellet insoluble material. Supernatants were transferred to new tubes. A portion of the sample was set aside to confirm optimal chromatin shearing by agarose gel electrophoresis.

### Preclear
Twenty µl of Dynabeads (Invitrogen) were blocked with 0.5 mg/ml BSA in PBS. Blocked Dynabeads were subsequently applied to each sonicated sample and rotated for 1 hr at 4°C. Samples were placed on a magnet stand for 1 min and precleared supernatant was transferred to a new tube; 5% of the sample was removed and stored at –80°C as input. Antibody and fresh 1× protease inhibitors were added to each sample. Samples were rotated overnight at 4°C.

### Pulldown
Samples were centrifuged at 14,000 rpm, 4°C, for 5 min to pellet insoluble material. Supernatants were transferred to new tubes; 50 µl of Dynabeads (Invitrogen) were blocked with BSA 5 mg/ml in PBS. Blocked Dynabeads were subsequently applied to each sample and rotated for 6 hr at 4°C.

### Stringency washes
All wash buffers were kept ice cold during stringency washes. Samples were washed eight times with RIPA Buffer (10 mM Tris-HCl pH 7.5, 140 mM NaCl, 1 mM EDTA, 0.5 mM EGTA, 1% Triton X-100, 0.1% SDS, 0.1% sodium deoxycholate, 2× protease inhibitors), two times with LiCl Buffer (10 mM Tris-HCl pH 8.0, 1 mM EDTA, 250 mM LiCl, 0.5% NP-40, 0.5% sodium deoxycholate, 2× protease inhibitors), two times with TE Buffer (10 mM Tris-HCl pH 8.0, 1 mM EDTA, 2× protease inhibitors).

### Elution and reversing crosslinks
One-hundred µl of Elution Buffer (10 mM Tris-HCl pH8.0, 5 mM EDTA, 300 mM NaCl, 0.1% SDS) was added to beads. Two µl of RNase A (Thermo Fisher, Cat#EN531) was added to each ChIP and input sample and incubated at 37°C for 30 min with gentle agitation. Ten µl of Proteinase K (Thermo Fisher, Cat# 25530049) was added to each sample and incubated at 37°C for 1 hr with gentle agitation. Crosslinks were reversed overnight at 65°C with gentle agitation. ChIP DNA was purified with a Qiagen MinElute PCR Purification kit (Cat#28004).

## ChIP-seq library construction and sequencing
ChIP-seq libraries were prepared using NEBNext ChIP-Seq Library Prep Reagent Set for Illumina (New England BioLabs, Cat# E6240). High-throughput sequencing was performed on Illumina HiSeq 2500 for single-end 50 bp reads or Illumina NovaSeq 6000 for paired-end 50 bp reads.

## ChIP-Rx-seq in zebrafish embryos
ChIP-Rx was adapted from *Orlando et al., 2014*, for H3K27me3 ChIP in zebrafish embryos treated with DMSO or PRT4165. Crosslinked *Drosophila melanogaster* S2 cells (ATCC Cat# CRL-1963) were spiked into resuspended zebrafish embryo pellets at a ratio of 5:1 (zebrafish cells: S2 cells). ChIP-Rx was subsequently performed in the same way as described above.

**Table 1.** Oligonucleotide sequences used for ChIP-qPCR for amplifying promoter regions.

| Target promoter | Direction | Sequence (5′→3′) |
| --- | --- | --- |
| *pax6a* | Forward | ctccggatccgaatcacaaaactagtcc |
| *pax6a* | Reverse | caaaggggtttgcaatctctcacaacc |
| *vsx1* | Forward | cccgtcatggtggcagtttc |
| *vsx1* | Reverse | gacagtgggatgatctgctggt |
| *isl1* | Forward | gtctcccatgtcaagaaagtaaggcg |
| *isl1* | Reverse | gccactttcccaccttcacagat |
| *idh3g* | Forward | cagcaagcgaacactgaccttgt |
| *idh3g* | Reverse | gcagttgggaaatacagcaaaggtacg |
| *pcf11* | Forward | cgatcgtttcagagcagccaataag |
| *pcf11* | Reverse | gtccgtcgtactttagcagagactg |
| *lman2* | Forward | cccgtccgttatatctgaatatacggaag |
| *lman2* | Reverse | ctcgtaaaatgccggtgtgtcac |

## ChIP in zebrafish sperm

ChIP in zebrafish sperm was conducted as described (*Murphy et al., 2018*).

qPCR was carried out using 2× SsoAdvanced Universal SYBR Green Supermix (Biorad Cat# 1725270) and a Biorad CFX real-time thermal cycler.

## Oligonucleotides

See *Table 1* for oligonucleotides used for ChIP-qPCR.

## RNA-seq

Total RNA was harvested from zebrafish embryos with a Qiagen Allprep kit (Cat# 80204). The Invitrogen DNA-*free* DNA removal kit (Cat# AM1906) was subsequently used to remove contaminating DNA from RNA samples. Intact poly(A) RNA was purified from total RNA samples (100–500 ng) with oligo(dT) magnetic beads and stranded mRNA sequencing libraries were prepared as described using the Illumina TruSeq Stranded mRNA Library Preparation Kit (RS-122–2101, RS-122–2102). Purified library quality was assessed on an Agilent Technologies 2200 TapeStation using a D1000 ScreenTape assay (Cat# 5067–5582 and 5067–5583). The molarity of adapter-modified molecules was defined by qPCR using the Kapa Biosystems Kapa Library Quant Kit (Cat# KK4824). Individual libraries were normalized to 5 nM and equal volumes were pooled in preparation for Illumina sequence analysis. High-throughput sequencing for RNAseq was performed on an Illumina HiSeq 2500. RNA-seq data displayed in *Figure 3—figure supplement 1A* was collected from http://www.ebiac.uk/gxa/experiments/E-ERAD-475. We would like to thank the Busch-Nentwich lab for providing RNA-seq data used in *Figure 3—figure supplement 1A*.

## ChIP-seq analysis

ChIP-seq Fastq files were aligned to Zv10 using Novocraft Novoalign with the following settings: -o SAM -r Random. SAM files were processed to BAM format, sorted, and indexed using Samtools (*Li et al., 2009*). ChIP-seq replica correlation was assessed with deepTools (*Ramírez et al., 2016*). Briefly, BAM files were read normalized with deeptools bamCoverage with the --normalizeUsingRPKM flag. Deeptools multiBigwigSummary bins and plotCorrelation were used to generate genome-wide correlation matrices for assessing replica correlation. ChIP-seq peak calling was accomplished using MACS2 with the following settings: callpeak -g 1.4e9 -B -q 0.01 -SPMR (*Zhang et al., 2008*). ChIP-seq peak calling for comparisons between Aebp2 and Jarid2 as well as ChIP-seq involving drug treatments was performed by utilizing the Multi-Replica Macs ChIPSeq Wrapper. Called peaks were annotated in R with the ChIPseeker package (*R Development Core Team, 2020*; *Yu et al., 2015*). Output bedgraph files from MACS2 were processed into bigwig files with UCSC Exe Utilities bedGraphToBigWig

(*Kent et al., 2010*). Resulting bigwig files were loaded into IGV for genome browser snapshots of ChIP-seq enrichment (*Thorvaldsdóttir et al., 2012*). Heatmaps of ChIP-seq enrichment at promoter and enhancer regions were made with deepTools (*Ramírez et al., 2016*). A bed file of Zebrafish zv10 UCSC RefSeq genes from the UCSC Table Browser was utilized for plotting heatmaps of ChIP enrichment at promoters (*Karolchik et al., 2004*). Genes residing on unmapped chromosomal contigs were excluded. Enhancer heatmaps utilized a bed file of postZGA H3K4me1 ChIP-seq (*Bogdanovic et al., 2012*) peak summits that had been filtered to exclude promoters and unmapped chromosomal contigs. Gene ontology analysis was performed with DAVID (*Huang et al., 2009*).

## Violin plot of ChIP-seq enrichment at promoters

Log2(ChIP/input) data for promoter regions (±1 Kb from TSS) of interest was collected from processed bigwig files by utilizing the program Bio-ToolBox 'get_datasets.pl'. Collected data was plotted in violin format. Unpaired t-tests with Welch's correction were utilized to determine statistical differences in ChIP enrichment between promoter K-means clusters. Violin plot and statistical analysis (*Figure 1—figure supplement 2J*) were performed in GraphPad Prism version 8.3.1 using GraphPad Prism version 8.3.1 for MacOS, GraphPad Software, San Diego, CA, https://www.graphpad.com.

## DNA motif analysis

HOMER was utilized for identifying putative TF-binding motifs present at promoters and enhancers (*Heinz et al., 2010*). The following parameters were used on bed files of promoters and enhancers of interest: findMotifsGenome.pl danRer10 -size –250,250. Known motifs (as opposed to de novo motifs) from HOMER were presented in *Figures 1 and 2*.

## ChIP-seq analysis involving drug treatments

Analysis of ChIP-seq experiments involving drug treatments was performed by utilizing the Multi-Replica Macs ChIPSeq Wrapper.

## RNA-seq analysis

RNA-seq fastq files were aligned to Zv10 using STAR (*Dobin et al., 2013*) with the following settings: --runMode alignReads --twopassMode Basic --alignIntronMax 50000 --outSAMtype BAM SortedBy-Coordinate --outWigType bedGraph --outWigStrand Unstranded --clip3pAdapterSeq AGATCGGA AGAGCACACGTCTGAACTCCAGTCA. The resulting sorted BAM files were subsequently indexed using Samtools (*Li et al., 2009*). FeatureCounts was utilized to collect count data for zv10 genes via the following command: -T 16s 2 –largestOverlap (*Liao et al., 2013*). Count data for all replicates across experimental conditions were combined into a single count matrix in R (*R Development Core Team, 2020*). This count matrix was subsequently used to identify differentially expressed genes with the R package DESeq (*Anders and Huber, 2010*). RNA-seq replica correlation was assessed with deepTools (*Ramírez et al., 2016*). Briefly, BAM files were read normalized with deeptools bamCoverage with the --normalizeUsingRPKM flag (*Ramírez et al., 2016*). Deeptools multiBigwigSummary bins and plotCorrelation were used to generate genome-wide correlation matrices for assessing replica correlation (*Ramírez et al., 2016*).

## Reprocessed ChIP-seq datasets

PreZGA H2Az and preZGA H3K4me1 ChIP-seq data (*Murphy et al., 2018*) (GEO: GSE95033), postZGA H3K4me1 ChIP-seq data (*Bogdanovic et al., 2012*) (GEO: GSE32483), preZGA and postZGA H3K27ac ChIP-seq data (*Zhang et al., 2018*) (GEO: GSE114954), postZGA H3K4me3 and H3K27me3 ChIP-seq data (*Zhang et al., 2014*) (GEO: GSE44269), and Nanog ChIP-seq (*Xu et al., 2012*) (GEO: GSE34683) were downloaded from the Gene Expression Omibus and reprocessed as described above.

## Whole genome bisulfite sequencing analysis

Whole genome bisulfite sequencing (WGBS) from *Potok et al., 2013* (DRA/SRA: SRP020008) was processed as described (*Murphy et al., 2018*).

## Data access

All sequencing datasets generated in this study have been deposited at the Gene Expression Omnibus under the accession number GSE168362.

## Drug treatments

PRT4165 (Tocris Cat#5047) was dissolved in DMSO at a concentration of 50 mM. PRT4165 was further diluted to a working concentration of 150 µM in embryo water and mixed vigorously. Zebrafish embryos were collected and immediately placed in embryo water containing 150 µM PRT4165 (or DMSO) and allowed to develop to the desired developmental stage.

## Acknowledgements

We thank Brian Dalley and the Huntsman Cancer Institute High-Throughput Sequencing Shared Resource for sequencing data. We thank Tim Parnell, Chongil Yi, and Jingtao Guo for helpful discussions regarding bioinformatic analysis. We thank David Grunwald and Rodney Stewart for providing helpful feedback during manuscript preparation. We thank all current and former members of the Cairns lab for their perspectives on this project. BRC is an investigator with the Howard Hughes Medical Institute. CW was funded by the T32 Developmental Biology Training Grant (59202072), 4DNucleome (NIH Common Fund) NBR – 92275293 S9001779, and HFSP RGP0025/2015. Imaging was performed at the University of Utah Microscopy Core (1S10RR024761-01). Financial support was received from the Howard Hughes Medical Institute and the Huntsman Cancer Institute core facilities (CA24014).

## Additional information

### Funding

| Funder | Grant reference number | Author |
| --- | --- | --- |
| Howard Hughes Medical Institute | Cairns | Bradley R Cairns |

The funders had no role in study design, data collection and interpretation, or the decision to submit the work for publication.

### Author contributions

Graham JM Hickey, Data curation, Formal analysis, Investigation, Methodology, Writing – original draft, Writing – review and editing; Candice L Wike, Formal analysis, Investigation, Methodology, Visualization, Writing – review and editing; Xichen Nie, Mengyao Tan, Investigation, Methodology; Yixuan Guo, Patrick J Murphy, Investigation, Methodology, Writing – review and editing; Bradley R Cairns, Conceptualization, Funding acquisition, Methodology, Project administration, Resources, Supervision, Writing – original draft, Writing – review and editing

### Author ORCIDs

Graham JM Hickey ⓘ http://orcid.org/0000-0001-9665-839X
Yixuan Guo ⓘ http://orcid.org/0000-0003-1940-1931
Bradley R Cairns ⓘ http://orcid.org/0000-0002-9864-8811

### Ethics

All of our work on zebrafish is authorized and overseen by our institutional animal care and use committee (IACUC). This work is authorized by the current IACUC Protocol #IACUC protocol 20-04011 'Germ Cell Epigenetics in Zebrafish'. We have no surgical procedures, and followed AVMA Guidelines involving a 2-step euthanasia for embryos and larvae which includes rapid chilling followed by immersion in a dilute sodium hypochlorite solution (discussed on page 89 of the 2020 Guidelines).

### Decision letter and Author response

Decision letter https://doi.org/10.7554/eLife.67738.sa1
Author response https://doi.org/10.7554/eLife.67738.sa2

# Additional files

## Supplementary files

• Transparent reporting form

## Data availability

Sequencing data have been deposited in GEO under accession code GSE168362.

The following dataset was generated:

| Author(s) | Year | Dataset title | Dataset URL | Database and Identifier |
|---|---|---|---|---|
| Hickey G, Cairns BR | 2021 | Establishment of Developmental Gene Silencing by Ordered Polycomb Complex Recruitment in Early Zebrafish Embryos | https://www.ncbi.nlm.nih.gov/geo/query/acc.cgi?acc=GSE168362 | NCBI Gene Expression Omnibus, GSE168362 |

The following previously published datasets were used:

| Author(s) | Year | Dataset title | Dataset URL | Database and Identifier |
|---|---|---|---|---|
| Murphy P, Cairns BR | 2017 | 'Placeholder' nucleosomes underlie germline-to-embryo DNA methylation reprogramming | https://www.ncbi.nlm.nih.gov/geo/query/acc.cgi?acc=GSE95033 | NCBI Gene Expression Omnibus, GSE95033 |
| Bogdanović O, Gómez-Skarmeta JL | 2012 | Dynamics of enhancer chromatin signatures mark the transition from pluripotency to cell specification during embryogenesis | https://www.ncbi.nlm.nih.gov/geo/query/acc.cgi?acc=gse32483 | NCBI Gene Expression Omnibus, GSE32483 |
| Zhang B, Xie W | 2018 | Widespread enhancer dememorization and promoter priming during parental-to-zygotic transition | https://www.ncbi.nlm.nih.gov/geo/query/acc.cgi?acc=GSE114954 | NCBI Gene Expression Omnibus, GSE114954 |
| Zhang Y, Vastenhouw NL, Liu XS | 2014 | Canonical Nucleosome Organization at Promoters Forms During Genome Activation | https://www.ncbi.nlm.nih.gov/geo/query/acc.cgi?acc=GSE44269 | NCBI Gene Expression Omnibus, GSE44269 |
| Xu C, Zon LI | 2012 | Nanog-like Regulates Endoderm Formation through the Mxtx2-Nodal Pathway | https://www.ncbi.nlm.nih.gov/geo/query/acc.cgi?acc=GSE34683 | NCBI Gene Expression Omnibus, GSE34683 |
| Murphy PJ, SF Wu, James CR, Wike CL, Cairns BR | 2013 | Reprogramming the Maternal Zebrafish Genome after Fertilization to Match the Paternal Methylation Pattern | https://www.ncbi.nlm.nih.gov/sra/?term=SRP020008 | NCBI Sequence Read Archive, SRP020008 |

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
