## [Editor Report]

This manuscript uses genomics tools and pharmacological treatment to study the chromatin landscape change in early-stage Zebrafish embryos, at the critical stage from using maternally deposited transcripts to actively turning on embryotic gene expression. In particular, this work addresses how key chromatin factors coordinate in regulating distinct groups of genes during early vertebrate development and should be of interest to researchers in chromatin biology and developmental biology fields.

---

## [Decision Letter]

**Decision letter after peer review:**

Thank you for submitting your article "Establishment of Developmental Gene Silencing by Ordered Polycomb Complex Recruitment in Early Zebrafish Embryos" for consideration by *eLife*. Your article has been reviewed by 3 peer reviewers, one of whom is a member of our Board of Reviewing Editors, and the evaluation has been overseen by Richard White as the Senior Editor. The following individual involved in review of your submission has agreed to reveal their identity: Charles K Kaufman (Reviewer #3).

Essential revisions:

1. This study first reported chromatin landscape changes in early zebrafish embryos at three distinct stages at preZGA, ZGA and postZGA stages, using a series of ChIP-seq experiments. This part of the results is of high quality, rigorously analyzed and clearly presented. The study then moved on and used a single drug to inhibit the enzyme with the hope to reduce ubH2A/Z mark. However, using the drug could lead to pleiotropic effect and cause general or secondary phenotypes. For example, this treatment leads to developmental arrest of embryogenesis at very early stage with low to no H3K27me3 mark, according to other data included in this work. Therefore, based on current results, it is hard to draw the conclusion for the causal connection from inhibiting RNF2 enzymatic function to the lack of ubH2A/Z, and subsequently to the lack of H3K27me3. In addition, due to the close interactions between these two Polycomb complexes (PRC1 and PRC2), the actual order could be reversed. Without a genetic experiment or more specific disruption assay, the conclusions should be carefully revised to be more balanced and reflect the caveats with the current results.

2. This sequential function of PRC1 and PRC2 has been reported in several other systems, including X inactivation and early embryogenesis in mice. The conceptual advance of this study could be extending this finding to another vertebrate system. However, due to the complications to draw conclusions (see #1 point), the Abstract and Discussion should be revised. Comparisons among these systems and research strategies should be further discussed.

3. H3K27me3 pattern should also be analyzed before ZGA. The authors detected a low level of H3K27me3 before ZGA using immunostaining. However, previous ChIP-seq analysis demonstrated that H3K27me3 are present on promoters before ZGA in zebrafish (e.g. PMID: 22137762).

4. It is unclear whether Aebp2 is the sole subunit for the recruitment of PRC2 to ubH2A/Z-marked regions. Paralleled analysis of the changes for Aebp2 and H3K27me3 upon RNF2 inhibitor treatment should be performed. And Aebp2-dependent and -independent regions should be separately classified for data analysis.

5. The role of PRC1 on the temporal regulation of gene expression needs to be better elucidated. The authors only analyzed the RNA-seq results for RNF2 inhibitor treated embryos postZGA. Therefore, it is currently not clear if the role of PRC1 in transcriptional repression is restricted to this stage. RNA-seq analysis of RNF2 inhibitor treated embryos on other stages are also warranted.

*Reviewer #2 (Recommendations for the authors):*

Some specific points:

Page 8, line 159-169. I'm confused by the authors' description of the enhancer cluster 5. By visual inspection, this cluster apparently only shared very moderate H3K4me1 and H3K27ac levels. The authors should provide a quantitative analysis similar to Figure 1S2J.

Figure 1 and 2. In Figure 1A and 2A. WGBS data of the preZGA stage embryo was shown. Yet in Figure 1B and 2B, the WGBS data was designated to postZGA embryo. Please confirm that the labeling is correct.

Page 10, line 209. Only a representative genomic region is shown in Figure 4S2B. Global analysis of the distribution changes for H3K27me3 needs to be performed in parallel with Aebp2 ChIP-seq analysis in Figure 4C.

Figure 3 Only colocalization was observed, yet functional links are not established in this study. Therefore, it is justified to claim that Aebp2 mediates H3K27me3 deposition at the current stage. Overall, the contribution of Jarid2 was not examined in the current stage.

*Reviewer #3 (Recommendations for the authors):*

To push the approach further, and not required for the current study, I wonder if another perturbation to test the model could be to delete key example genomic loci (or, perhaps even better, mutate Figure 1E candidate TF binding sites in the region?) to test for loss of proper Placeholder histone deposition. Or, engineer transgenic fish with developmental enhancer/promoter sequences in an ectopic location (or profile an existing transgenic around the time of ZGA) to show that they can drive establishment of the set of Placeholder chromatin/DNA marks that, in their model, delineate developmental gene loci that are silenced at the ZGA but will later be expressed.

---

## [Author Response]

Essential revisions:1. This study first reported chromatin landscape changes in early zebrafish embryos at three distinct stages at preZGA, ZGA and postZGA stages, using a series of ChIP-seq experiments. This part of the results is of high quality, rigorously analyzed and clearly presented. The study then moved on and used a single drug to inhibit the enzyme with the hope to reduce ubH2A/Z mark. However, using the drug could lead to pleiotropic effect and cause general or secondary phenotypes. For example, this treatment leads to developmental arrest of embryogenesis at very early stage with low to no H3K27me3 mark, according to other data included in this work. Therefore, based on current results, it is hard to draw the conclusion for the causal connection from inhibiting RNF2 enzymatic function to the lack of ubH2A/Z, and subsequently to the lack of H3K27me3. In addition, due to the close interactions between these two Polycomb complexes (PRC1 and PRC2), the actual order could be reversed. Without a genetic experiment or more specific disruption assay, the conclusions should be carefully revised to be more balanced and reflect the caveats with the current results.

We thank the reviewers for their overall positive view of the work and examination of the chromatin landscape changes via ChIP approaches. The issues raised here include: 1) whether the inhibition of Rnf2 is appropriate functional evidence, and 2) whether we are confident in the order of H2Aub1 (PRC1 action) preceding H3K27me3 (PRC2 action).

First, regarding our inhibition experiments – our work, like the other two papers published in *Nature Genetics* (Mei et al., 2021; Chen et al., 2021), each used one approach to lower H2Aub1. We agree that our approach could have off target effects, and now state that in the Discussion. We note, however, that the other two mouse papers likewise each took one approach to lower H2Aub1: one employed BAP overexpression, which arrested embryos at the 4-cell stage (when H3K27me3 is extremely low) and therefore could not assess whether H3K27me3 was impaired when H2Aub1 was lowered. The second employed maternal, but not paternal, genetic disruption of PRC1, which showed a transient lowering of H2Aub1 and H3K27me3, but restoration by the morula stage. Therefore, each approach has its advantages and limitations, which we now discuss in the revised Discussion. However, together, all three papers provide complementary evidence for the causal relationship.

We respectfully disagree that current evidence supports the notion that “the actual order (of PRC1/2 complexes) could be reversed”. Our evidence of PRC1/H2Aub1 preceding PRC2/H3K27me3 does not rely solely on the PRT4165 inhibitor experiment – it is fundamentally supported by the temporal relationships of the modifications. In preZGA embryos we clearly detect H2Aub1 by western and IF approaches, and importantly detect robust ChIP specifically at thousands of developmental genes. In contrast, we and almost all other investigators detect no H3K27me3 by Western or IF, nor can we and almost all others detect it at any genes. As detailed in point #3 below – the one paper that has claimed preZGA H3K27me3 (Lindemann et al., 2011) showed low H3K27me3 at only 83 genes during preZGA through ZGA, not at the thousands of developmental genes that bear H2Aub1 during preZGA through ZGA. Thus, the vast majority of the evidence points strongly to H3K27me3 being virtually absent during preZGA, whereas we show the clear presence of H2Aub1, and its residence at large numbers of developmental genes during preZGA.

We will note here (and briefly in the Discussion) that the lack of H3K27me3 has no measurable effect on the progression of zebrafish embryos through gastrulation, and through 24hpf, at which they display a normal body plan and organogenesis (though fail by 48hrs). This was determined in work published by L. Kamminga and colleagues (Rougeot et al., Development 2019; on which we were co-authors) using maternal zygotic *ezh2* null mutations. Thus, H3K27me3 appears to play only minor roles until late stages of development, where it acts in a canonical manner.

We agree with the reviewers that a complementary approach would help to validate our results with the inhibitor, and we expended significant effort on this issue during revision. Regarding alternative inhibitors, PRT4165 is currently the only commercially available inhibitor of Rnf2. In the absence of alternative inhibitors, we attempted genetic approaches, beginning with the expression of a minimal version of the H2A deubiquitinase complex, PR-DUB in early zebrafish embryos. Here, single-cell stage zebrafish embryos were microinjected with mRNA encoding the two essential subunits of PR-DUB, *bap1* and *asxl2*, to disrupt H2A/Zub1. As a negative control we utilized a point mutant of *bap1* that abolishes catalytic activity. Injected embryos were allowed to develop to 4hpf and subsequently subjected to western blot analysis to assay H2Aub1 abundance. Unfortunately, this approach yielded no appreciable change in bulk levels of H2Aub1 compared to control, despite attempting multiple different doses of the *bap1* and *asxl2* mRNAs. Importantly, we validated (by an installed V5 tag) by Western analysis that the proteins were each expressed. Here, we reason that additional factors (beyond the two we expressed) might be needed to create a fully active (or properly targeted) BAP/ASXL2 complex and/or that inhibitors of BAP1 might be present that prevented enzymatic activity. Regardless, the inability to greatly lower H2A/Zub1 precluded further analyses.

As a tertiary approach to lower H2A/Zub1, we micro-injected single-cell stage zebrafish embryos with a mutant *h2af/z* mRNA encoding and form of H2A.Z that cannot be ubiquitinated. We also injected wild-type *h2af/z* as a negative control. Unfortunately, our Western analyses showed that we were not able to arrive at protein levels derived from the injected species that approached the large stores of WT H2A.Z protein present in the embryo, so competition was not achieved, and H2A/Zub1 was not lowered.

Taken together, our work parallels the work earlier this year from two groups who each took one effective approach to lower H2AUb1: ours with PRT4165 and theirs with BAP1/ASXL1 overexpression or a PRC1 maternal-specific mutant approach. Each approach has advantages and caveats, and the reviewer rightly points out that our approach can have off target affects. Therefore, we have included this caveat and how it may affect our interpretations accordingly in the revised paper (see Discussion, and also point #2, below). Specifically, in the Discussion of the revised manuscript we state:

“However, it is important to note that PRT4165 may have off-target effects aside from inhibition of Rnf2 which may contribute to our observations.” (page 16, line #336-337)

and

“However, as PRT4165 was the only approach we employed that successfully removed the vast majority of H2Aub1, future studies involving orthogonal approaches to elicit Rnf2 loss in zebrafish will be needed to further validate the consequences of H2Aub1 loss in early zebrafish embryos” (page 16 and line #342-345).

2. This sequential function of PRC1 and PRC2 has been reported in several other systems, including X inactivation and early embryogenesis in mice. The conceptual advance of this study could be extending this finding to another vertebrate system. However, due to the complications to draw conclusions (see #1 point), the Abstract and Discussion should be revised. Comparisons among these systems and research strategies should be further discussed.

There is indeed precedent for that order in X chromosome inactivation in ES cells, and we have cited that work in the revision. In addition, the work published earlier this year in *Nature Genetics* that determines the order for the X chromosome and developmental genes in mouse embryos is now discussed in the revised version (pages 16-18, lines 346-383, and see below). However, at the time when we submitted our work, the order of polycomb complex action at developmental genes was not known. Our manuscript was submitted and progressed to full review at *eLife* (and was posted on the BioRxiv) prior to the work in mice being published in *Nature Genetics*. Therefore, we hope that the novelty and conceptual advance of the work will be assessed based on our date of submission.

The recent mouse work (in *Nature Genetics*) on H2AUb inhibition in mice strongly supports our work in zebrafish, and we have now included those papers in the Discussion of our revised paper (pages 16-18, lines 346-383). Indeed, their methods for reducing H2AUb1 in mouse embryos led to upregulation of developmental genes, in keeping with our observations. Regarding novelty, our work goes much further, as only our paper addresses the mechanism by which H2AUb1 is ‘read’ in the early embryo by a non-canonical PRC2 complex (bearing Aebp2) to place H3K27me3 on developmental genes. Therefore, we believe our work has conceptual advances both in the areas of overlap, and also in unique aspects related to PRC2 targeting.

3. H3K27me3 pattern should also be analyzed before ZGA. The authors detected a low level of H3K27me3 before ZGA using immunostaining. However, previous ChIP-seq analysis demonstrated that H3K27me3 are present on promoters before ZGA in zebrafish (e.g. PMID: 22137762).

The paper referred to (Lindeman et al., 2011; PMID 22137762) is an interesting and influential paper in the field, however some of the interpretations have required revisiting. The Lindeman paper does claim that H3K27me3 can be found on developmental promotes during preZGA. However, the key questions are – how many promoters, what fraction of promoters, and how robust is the marking? The Lindeman paper and follow-up work by us and others (Zhang et al., *Genome Research* 2014; Zhang et al., *Molecular Cell*, 2018), agree that there are >3500 developmental genes marked by H3K27me3 in postZGA. However, the Lindeman data claims that only 201 genes are marked by H3K27me3 in preZGA embryos (DocumentS2, Table S1; Venn Diagrams in Lindeman et al., 2011; PMID 22137762). Also, one expects that if the purpose of that H3K27me3 involves gene poising, it should be stable and last into postZGA – and when one queries their data and asks how many genes bear H3K27me3 from preZGA through postZGA, the number drops to 83 genes. These 83 genes are extremely few compared to the >3,500 genes marked in postZGA embryos by H3K27me3 in the Lindeman data and Zhang data. Also, these 83 H3K27me3 genes show very low median ChIP signal during preZGA (less than 2-fold enrichment) and only show clear (>4-fold enrichment) during postZGA (see Author response image 1).

**Author response image 1. sa2fig1:** 

Importantly, we emphasize that H3K27me3 ChIP experiments from the Schier lab (Vastenheow et al., *Nature*, 2010), Wei Xie’s lab (Zhang et al., *Molecular Cell*, 2018) and our lab (described as ‘data not shown’ in Murphy et al., *Cell* 2018) did not reveal any peaks for H3K27me3 during preZGA. Furthermore, both our IF data (this work; Figure 1, FigureS1, Panel B) and Zhang et al., 2018 (Supplemental Figure S1A) fail to show H3K27me3 signal preZGA. Taken together, work from several labs supports the view that there is extremely low H3K27me3 in the genome at preZGA, which has resulted either in a lack of ChIP peaks preZGA (Schier, Xie and Cairns labs) or detection of very low H3K27me3 at a very small number of loci (~83-201; Lindeman et al). However, all groups agree that after ZGA, H3K27me3 increases dramatically, enabling clear detection at >3500 genes. Therefore, the much larger questions – addressed in our work – are how >3500 developmental genes are silenced at ZGA (we show it is H2AUb1, not H3K27me3), and how those >3500 obtain high H3K27me3 (we show it is by non-canonical PRC2 binding to H2AUb1).

4. It is unclear whether Aebp2 is the sole subunit for the recruitment of PRC2 to ubH2A/Z-marked regions. Paralleled analysis of the changes for Aebp2 and H3K27me3 upon RNF2 inhibitor treatment should be performed. And Aebp2-dependent and -independent regions should be separately classified for data analysis.

We thank the reviewers for raising these important points, which we have addressed. Previous data including the very nice recent cryo-EM structure of the non-canonical PRC2 complex bound to H2Aub1+ nucleosomes from Eva Nogales’ group (Kasinath et al., *Science* 2021), indicates that Aebp2 is not the sole subunit involved in recruitment of PRC2 to H2A/Zub1. Rather, Aebp2 and Jarid2 subunits of ncPRC2 both contribute to recruitment of PRC2 via direct contacts with H2A/Zub1. However, we note that the PRC2 complex can be isolated from cells or produced in a recombinant form with Aebp2, but lacking Jarid2 (Cao et al., *Science* 2002; Ciferri et al., *eLife* 2012) – suggesting that these proteins are not together in all PRC2 complexes, and that Aebp2 assembles well into PRC2 without Jarid2. Accordingly, we have modified the text of our manuscript to clarify that Aebp2 is not the sole PRC2 subunit involved in recruitment to H2A/Zub1+ nucleosomes.

To investigate the overlap between Aebp2 and Jarid2 concept further, we performed ChIP-seq of Jarid2 in postZGA embryos. We have included our Jarid2 ChIP profiling into our manuscript in Figure 3 —figure supplement 2. Although the ChIP signal and dynamic range were modest, the majority of Jarid2 sites overlap with Aebp2-bound sites, supporting the notion that these two proteins may work together at a portion of PRC2-bound sites. However, given the modest ChIP signal of Jarid2, we do not believe that highly quantitative statements about binding selectivity can be made.

Next, as requested – we reanalyzed our ChIP-seq data for Aebp2 and H3K27me3 upon Rnf2 inhibition in a paralleled manner. This was an important addition to the paper. This paired Aebp2/H3K27me3 analysis has been incorporated into new Figure 4 and reflected in the text of the manuscript. The clear result is that the same places that lose Aebp2 binding (following the loss of H2AUb1) also lose H3K27me3. We believe this revised way of presenting the data conveys the result much clearer than the previous version of Figure 4.

5. The role of PRC1 on the temporal regulation of gene expression needs to be better elucidated. The authors only analyzed the RNA-seq results for RNF2 inhibitor treated embryos postZGA. Therefore, it is currently not clear if the role of PRC1 in transcriptional repression is restricted to this stage. RNA-seq analysis of RNF2 inhibitor treated embryos on other stages are also warranted.

The reviewers raise an interesting question, which also addresses whether the removal of H2AUb from developmental genes might enable their activation during preZGA, which would confer to developmental genes the status/privilege of ‘minor wave’ genes. To address this question, we performed RNA-seq at preZGA and ZGA timepoints with embryos treated with the RNF2 inhibitor (PRT4165) or vehicle control (DMSO) and examined whether precocious transcription of developmental genes occurred prior to post ZGA. This work has been incorporated into our revised manuscript in Figure 4 —figure supplement 3. GO-term analysis of genes meeting our threshold criteria (p-value ≤0.01, fold change ≥1.5) uncovered no enrichment of developmental gene activation upon RNF2 inhibition prior to postZGA. Instead, enriched GO-terms of upregulated transcripts correspond to RNA-binding and ribosomal protein genes. Thus, chromatin de-repression (via H2Aub1 removal) does not cause precocious transcriptional activation of developmental genes until postZGA (4hpf).

Here, we reason that the observed precocious transcription of developmental genes at postZGA in embryos treated with PRT4165 relies on binding of opportunistic transcriptional activators. However, these activators may be temporally constrained during preZGA by: 1) translation of mRNA encoding these factor(s) and/or 2) chromatin remodeling events needed to expose binding sites for opportunistic activators.

Reviewer #2 (Recommendations for the authors):Some specific points:Page 8, line 159-169. I'm confused by the authors' description of the enhancer cluster 5. By visual inspection, this cluster apparently only shared very moderate H3K4me1 and H3K27ac levels. The authors should provide a quantitative analysis similar to Figure 1S2J.

We apologize for the confusion, which also raised a comment from Reviewer #1. There are two points of clarification to make here. First, we now clarify in the manuscript that DNAme patterns are essentially identical/static from 2.5hpf (preZGA) to 4.3hpf (postZGA). On Page 5, line 100 we now state:

“For our comparisons to DNA methylation (DNAme), we note that DNAme patterns are reprogrammed between fertilization and the preZGA (2.5hpf) timepoint (Potok et al., 2013; Jiang et al., 2013), but remain static in zebrafish embryos from 2.5 hpf (preZGA) to 4.3 hpf (postZGA). Therefore, for brevity we chose to display only a single timepoint for DNAme data in subsequent figures, which is representative of all developmental stages examined by the genomics approaches in this work”.

Second, our reference to H3K27ac being very high refers specifically to enhancer cluster 5 in the postZGA state. Thus, H3K27me3 is very high, and these loci are DNA methylated (DNAme shown for preZGA, but identical/methylated in postZGA).

Figure 1 and 2. In Figure 1Aand2A. WGBS data of the preZGA stage embryo was shown. Yet in Figure 1B and 2B, the WGBS data was designated to postZGA embryo. Please confirm that the labeling is correct.

We thank the reviewer for their comment. The labeling is correct. As noted above, the DNA methylation state remains constant at enhancers and promoters from preZGA to postZGA.

Page 10, line 209. Only a representative genomic region is shown in Figure 4S2B. Global analysis of the distribution changes for H3K27me3 needs to be performed in parallel with Aebp2 ChIP-seq analysis in Figure 4C.

We agree and have addressed: Figure 4 has been largely revised to examine the changes in localization of Aebp2 and H3K27me3 in parallel, as also requested in the list of unified comments.

Figure 3 Only colocalization was observed, yet functional links are not established in this study. Therefore, it is justified to claim that Aebp2 mediates H3K27me3 deposition at the current stage. Overall, the contribution of Jarid2 was not examined in the current stage.

Here, our functional link is that when H2Aub1 is prevented (via PRT4165), Aebp2 fails to occupy all sites that formerly had H2Aub1, and H3K27me3 is not deposited at those same loci – which is now much more clearly displayed through the revised analyses present in Figure 4.

The reviewer raises an interesting issue in their request to examine Jarid2. Here, although the Jarid2 antibody and resulting ChIP signal was not as robust as with our potent Aebp2 antibody (limiting highly quantitative conclusions) the majority of Jarid2 targets were also Aebp2 targets – consistent with Aebp2 and Jarid2 working together for H2Aub1 recognition (Figure 3 —figure supplement 2).

Reviewer #3 (Recommendations for the authors):To push the approach further, and not required for the current study, I wonder if another perturbation to test the model could be to delete key example genomic loci (or, perhaps even better, mutate Figure 1E candidate TF binding sites in the region?) to test for loss of proper Placeholder histone deposition. Or, engineer transgenic fish with developmental enhancer/promoter sequences in an ectopic location (or profile an existing transgenic around the time of ZGA) to show that they can drive establishment of the set of Placeholder chromatin/DNA marks that, in their model, delineate developmental gene loci that are silenced at the ZGA but will later be expressed.

We appreciate the reviewer’s insightful comments. In the future we will use Crispr-CAS9 mediated targeting of transcription factors and histone modifications to test these interesting questions.